

# Impact of anthropogenic and biogenic sources on the seasonal variation of the molecular composition of urban organic aerosols: a field and laboratory study using ultra-high resolution mass spectrometry

Kaspar R. Daellenbach[1,*], Ivan Kourtchev[2], Alexander L. Vogel[1,**], Emily A. Bruns[1], Jianhui Jiang[1], Tuukka Petäjä[3], Jean-Luc Jaffrezo[4], Sebnem Aksoyoglu[1], Markus Kalberer[2,***], Urs Baltensperger[1], Imad El Haddad[1], André S. H. Prévôt[1]

[1]Laboratory of Atmospheric Chemistry, Paul Scherrer Institute (PSI), 5232 Villigen-PSI, Switzerland
[2]Department of Chemistry, University of Cambridge, Cambridge, U. K.
[3]Institute for Atmospheric and Earth System Research / Physics, Faculty of Science, University of Helsinki, Helsinki, Finland
[4]Université Grenoble Alpes, CNRS, IRD, Grenoble INP, IGE, Grenoble, France
[*]now at: Institute for Atmospheric and Earth System Research / Physics, Faculty of Science, University of Helsinki, Helsinki, Finland
[**]now at: Institute for Atmospheric and Environmental Sciences, Goethe University, Frankfurt am Main, Germany
[***]now at: Department of Environmental Sciences, University of Basel, Basel, Switzerland

*Correspondence to*: André S. H. Prévôt (andre.prevot@psi.ch) and Markus Kalberer (markus.kalberer@unibas.ch)

**Abstract.**

This study presents the molecular composition of OA using ultra-high resolution mass spectrometry (Orbitrap) at an urban site
in Central Europe (Zurich, Switzerland). Specific source spectra were also analysed, including samples representative of wood burning emissions from Alpine valleys during wood burning pollution episodes and chamber investigations of wood smoke as well as samples from Hyytiälä strongly influenced by biogenic secondary organic aerosol. While samples collected during winter in Alpine valleys have a molecular composition remarkably similar to fresh laboratory wood burning emissions, winter samples from Zurich are influenced by more aged wood burning emissions. In addition, other organic aerosol emission or
formation pathways seem to be important at the latter location in winter. Samples from Zurich during summer are similar to those collected in Hyytiälä, predominantly impacted by oxygenated compounds with an H/C ratio of 1.5, indicating the importance of biogenic precursors for SOA formation at this location. We could explain the strong seasonality of the molecular composition at a typical European site by primary and aged wood burning emissions and biogenic secondary organic aerosol formation during winter and summer, respectively. Results presented here likely explain the seasonally rather constant
predominance of non-fossil organic carbon at European locations.



# 1 Introduction

Aerosols affect earth's climate, ecosystems, and human health. A main contributor to the aerosol mass is the organic aerosol (OA). OA can be directly emitted as primary particles (POA) or produced by oxidation and subsequent condensation of volatile organic compounds (VOC) (secondary OA, SOA). Sources of both POA and SOA can be natural such as plant debris, resuspension and biogenic VOC oxidation, or anthropogenic from traffic, cooking, and residential heating using wood or fossil fuels. The resulting SOA is typically a highly complex mixture of unknown compounds, the chemical characterization of which requires comprehensive analytical strategies (e.g., Noziere et al., 2015).

Field-deployments of the aerosol mass spectrometer (AMS, Canagaratna et al., 2007) at several European stations have revealed a large impact of primary wood burning emissions on OA (e.g. between 11-59% in Switzerland ), while primary traffic emissions have a smaller contribution (4-14%) (Lanz et al., 2010; Gilardoni et al., 2011; Daellenbach et al., 2017). The results are consistent with radiocarbon measurements (Zotter et al., 2014a), showing that during extreme winter pollution episodes non-fossil organic carbon (OCnf) may account for up to 97% of OC at Alpine valley sites (Magadino: 83%, S. Vittore: 97%) and 74% at an urban background site in Zurich and are associated with levoglucosan, a pyrolysis product of cellulose. Similar results were obtained in French Alpine valleys such as in Chamonix (Bonvalot et al., 2016). Based on AMS measurements, a large fraction of OA, in addition to primary wood burning emissions, is found to be SOA from unknown origins. Radiocarbon analyses suggest that a large fraction of SOA is also non-fossil, potentially arising from biomass smoke aging (Vlachou et al., 2018). Recent smog chamber studies were capable of clearly showing the significant SOA formation rates during the aging of wood burning emissions, which could be largely attributed to phenolic derivatives together with benzene and naphthalene (Bruns et al., 2016).During summer the dominant fraction of OA was reported to be SOA (Bozzetti et al., 2016; 2017a; 2017b; Canonaco et al., 2015; Daellenbach et al., 2017; Lanz et al., 2010; Reyes-Villegas et al., 2016; Schlag et al., 2016), at many European sites, even in large industrialized cities such as Marseilles, France (El Haddad et al., 2011). Based on radiocarbon analyses, it is hypothesized that summertime SOA in Central Europe is largely produced from the oxidation of biogenic precursors (Vlachou et al., 2018; Bonvalot et al., 2016; Zotter et al., 2014a). While this is found to be the case in Europe, a strong contribution of fossil fuel emissions to SOA has been observed in the Los Angeles basin (Zotter et al., 2014b; Platt at el., 2017).

Despite these recent advancements of OA source apportionment using the AMS and radiocarbon analysis, direct links between observed SOA and its precursors are still missing. This remaining gap is strongly related with the extensive molecular fragmentation in the AMS caused by the use of electron ionization, which hinders retrieving detailed information on the chemical nature of winter and summertime SOA fractions. The use of soft ionization techniques, such as electrospray ionization coupled to ultra-high-resolution mass spectrometry (ESI-UHR-MS) is a promising technique that may help bridging such existing gaps (Nizkorodov et al., 2011). The technique provides with minimal fragmentation the exact mass of molecular ions,



thus allowing the determination of the SOA molecular composition. The disadvantage of the technique is the relatively strong variability in the ionization efficiency of different compounds; i.e. the relative contribution of a compound cannot be directly linked to its concentration without using an authentic standard (Huffmann et al., 2012; Kruve et al., 2017; Noziere et al., 2015). Despite this, laboratory studies using ESI-UHR-MS have deepened our understanding of the formation of SOA from various

sources such as isoprene, monoterpenes and vehicular exhaust (Bateman et al., 2009; Kourtchev et al., 2015; Mutzel et al., 2015; Nguyen et al., 2010, 2011; Romonosky et al., 2017; Walser et al., 2008). Studies focusing on the ESI-UHR-MS analysis of aerosol samples collected in field campaigns qualitatively revealed the influence of different sources at various sites (Dzepina et al., 2015; Kourtchev et al., 2013, 2014b; Lin et al., 2012; O'Brien et al., 2013; 2014; Rincón et al., 2012; Roach et al., 2010; Tao et al., 2014; Tong et al., 2016; Wang et al., 2017). However, the typical number of samples analyzed remains

very limited due to extensive work required both in the laboratory as well as for data analysis. As a result, only limited knowledge is currently available on the changes of OA molecular characteristics throughout the seasons and especially during winter.

In this study, we examine the seasonal variability of the OA chemical composition at a molecular level, at an urban background

site in Central Europe (Zurich, Switzerland). In order to elucidate the influence of different sources on SOA chemical compositions, samples from Zurich are compared to those collected during wood burning episodes and wood burning smog chamber experiments, as well as samples dominated by biogenic SOA.

## 2 Experiments & Methods

### 2.1 Aerosol sample collection

$PM_{10}$ quartz fiber filter daily samples were collected at 3 sites in Switzerland in 2013 using a High Volume samplers (500 l min$^{-1}$, 14.7 cm filter diameter, 24 h, 15 samples in Zurich, 4 samples in Magadino, and 4 samples in S. Vittore, Fig. S1). Zurich is located on the northern Swiss plateau and the site is classified as urban background. The sites in Magadino and S. Vittore are in Alpine valleys in Southern Switzerland.

Additional samples impacted by specific sources were also collected. The SMEAR-II station (Station for Measuring Ecosystem Atmosphere Relations, Hari and Kulmala 2005) at Hyytiälä is a rural background site in Finland, strongly influenced by biogenic SOA (vegetation dominated by Scots pine and Norway spruce). The $PM_1$ aerosol was collected at the SMEAR-II station between 16 and 25 August 2011 and between 7 July and 4 August 2014 using a low volume sampler (35 l min$^{-1}$) (Kourtchev et al., 2016). By including these samples, it was possible to account for inter-annual variability arising from

temperature differences.



Smog chamber experiments were performed to examine the composition of wood burning emissions from the stable flaming phase and their evolution with aging (Bruns et al., 2016; 2017). Fresh emissions were first injected into a 6 m$^3$ Teflon smog chamber (Platt et al., 2013; Bruns et al., 2016, 2017). After 30 min of mixing, particles were sampled onto filters (UV-lights off, sampling time 30 mins at ~30 l/min). Then, emissions were photochemically aged in the smog chamber, by injecting

HONO at a flow rate of 1-2 l/min, which generates OH radicals upon photolysis. Filter samples were collected at equivalent atmospheric aging times of 10 and 30 h determined by the Barmet et al. (2012) method, assuming a winter time OH concentration of 10$^6$ molec cm$^{-3}$.

## 2.2 Direct-injection (-)ESI-UHR-MS

### 2.2.1 Analytical procedure

For each sample, a part of the quartz fiber filter was extracted three times with 5 mL of methanol (Optima® grade, Fisher Scientific) under ultrasonic agitation in an ice chilled bath for 30 min. The three extracts were combined, filtered through a Teflon® filter (0.2 μm) and reduced by volume to approximately 200 μL under a gentle stream of nitrogen. Extracts of field blanks were prepared analogously.

The aerosol extracts were analyzed using an ultra-high resolution LTQ Orbitrap Velos mass spectrometer in the negative mode (Thermo Fisher, Bremen, Germany) equipped with a TriVersa Nanomate robotic nanoflow chip-based ESI source (Advion Biosciences, Ithaca NY, USA). The analyses were conducted with an ionization voltage of 1.51 kV, a vaporizer temperature of 230°C for the Zurich samples and 254°C for the other samples, and a capillary temperature of 200°C. The Orbitrap MS instrument was calibrated using an Ultramark 1621 solution (Sigma-Aldrich, UK). The accuracy of the *m/z* calibration was <2

ppm. The mass resolution of the instrument was 100'000 at *m/z* 400. Two or three replicate measurements were conducted for each extract, and field blank extracts were analyzed in the same way.

### 2.2.2 Data analysis

For each replicate measurement, approximately 38 mass spectral scans were averaged into 1 mass spectrum per measurement (representing 1 min measurement time). The gathered mass spectral information was processed with Xcalibur 2.1 (Thermo

Scientific). In a first step, peaks exhibiting a signal below instrumental noise level were removed. We estimated the instrumental noise level as the 99.9$^{th}$ percentile of the signal recorded in regions of the mass spectrum where no signal is expected. In a second step, for all remaining peaks possible molecular compositions were assigned to the signals using a tolerance level within ±5 ppm and the following constraints: $^{12}$C≤100, $^{13}$C≤1, $^{1}$H≤200, $^{16}$O≤50, $^{14}$N≤5, $^{32}$S≤2 and $^{34}$S≤1. Among all the assigned molecular compositions, only molecular assignments in agreement with the following criteria described in

detail in Kourtchev et al. (2013) were further considered.



1. Molecular assignments of a peak have to be within a defined mass difference from the measured mass (*m/z*). The tolerated difference between the molecular assignments and the measured mass of a peak was assessed based on the respective difference for nine known compounds for every analyzed sample (on average 0.5 ppm).

2. Molecular assignments have to be in agreement with the following elemental ratios: i) O/C≤1.5, ii) 0.3≤H/C≤2.5, iii) 0≤N/C≤0.5, iv) 0≤S/C≤0.2.

3. Molecular assignments have to be consistent with a neutral formula with a positive integer double bond equivalent, (DBE, for any chemical formula $C_{N_C}H_{N_H}O_{N_O}N_{N_N}S_{N_S}$):

$$DBE = 1 - \frac{N_H}{2} + \frac{N_N}{2} + N_C \qquad (1)$$

4. Nitrogen containing compounds have to be in agreement with the nitrogen rule.

5. Formulae including either or both [13]C and [34]S were only further considered in the presence of a counterpart including only [12]C and [32]S, respectively.

6. Only peaks were further considered if their intensity in the sample was at least 3 times larger than in the blank.

## 2.2.3 Computation of bulk properties

All properties, molar ratios, and chemical formulae presented in this manuscript refer to neutral molecules. Literature data was additionally also filtered with criterion (2) for comparability. Bulk elemental ratios (H/C, O/C, N/C and S/C) and the number of carbons of the organic aerosol were computed as follows (Nizkorodov et al., 2011, Bateman et al., 2012):

$$(O/C)_{bulk} = \sum_i x_i * N_{O,i} / \sum_i x_i * N_{C,i} \qquad (2)$$

$$(H/C)_{bulk} = \sum_i x_i * N_{H,i} / \sum_i x_i * N_{C,i} \qquad (3)$$

$$(N/C)_{bulk} = \sum_i x_i * N_{N,i} / \sum_i x_i * N_{C,i} \qquad (4)$$

$$(S/C)_{bulk} = \sum_i x_i * N_{S,i} / \sum_i x_i * N_{C,i} \qquad (5)$$

$$C_{bulk} = \sum_i x_i * N_{C,i} / \sum_i x_i \qquad (6)$$

O/C, H/C, N/C, S/C, and carbon number probability distribution functions (pdf) are weighing the number of O, H, N, S, C atoms in a compound $i$ ($N_{O,i}$, $N_{H,i}$, $N_{N,i}$, $N_{S,i}$, $N_{C,i}$,) by the respective peak intensity $x_i$ such that the probability $p$ of, e.g., an H/C of $h$ is defined as:

$$p(H/C = h) = \sum_i x_{i|H/C=h} / \sum_i x_i \qquad (7)$$



The compounds' aromaticity can be estimated by different parameterizations (e.g., aromaticity index, AI, and aromaticity equivalent, $X_c$). $X_c$ (Yassine et al., 2014) has the advantage over AI (Koch and Dittmar, 2006) that (poly-)aromatic compounds with significant alkylation are accurately classified. Therefore, we apply the widely used aromaticity equivalent ($Xc$) in this study, expressed as follows:

$$X_c = \frac{3*(DBE-(m*N_O+n*N_S))-2}{DBE-(m*N_O+n*N_S)} \tag{8}$$

Here, DBE is the double bond equivalent (Eq. 1), $N_O$, $N_S$ are the number of oxygen and sulfur atoms present in the molecule, respectively, and $m$ and $n$ are the number of oxygen and sulfur atoms involved in π-bonds, respectively (both assumed to be
0.5, Yassine et al., 2014). For molecular formulae with an odd number of sulfur or oxygen, the sum $(m*N_O + n*N_S)$ was rounded down to the closest integer, and for compounds with $DBE \leq m*O + n*S$ $X_c$ was set to 0 (Yassine et al., 2014). Compounds with $X_c<2.5$ were considered non-aromatic, $X_c\geq2.5$ aromatic, and $X_c\geq2.7143$ condensed aromatic (Yassine et al., 2014).

## 2.3 Other chemical analyses

The chemical analyses were complemented with other filter-based analyses. Organic and elemental carbon (OC, EC) concentrations were determined for Zurich, Magadino, S. Vittore and Hyytiälä by a thermo-optical transmission (TOT) method with a Sunset OC/EC analyzer (Birch and Cary, 1996), following the EUSAAR-2 thermal-optical transmission protocol (Cavalli et al., 2010). Water-soluble inorganic ions ($K^+$, $Na^+$, $Mg^{2+}$, $Ca^{2+}$, $NH_4^+$; and $SO_4^{2-}$, $NO_3^-$, $Cl^-$) were measured by ion chromatography for Zurich, Magadino and S. Vittore (Piazzalunga et al., 2013 and Jaffrezo et al., 1998). Pinic acid and 3-
MBTCA were quantified for the ambient filter samples from Switzerland, with analysis by LC-MS ((-)ESI- LCQ-Fleet, Thermos-Fisher), with chromatographic separation performed on a Synergi 4μm Fusion – RP 80A (Phenomenex) with a water – acetonitrile – formic acid eluent. The calibration was performed with authentic standards. On-line measurements of gas-phase compounds (NOx by chemiluminescence, $SO_2$ by fluorescence/absorption) were performed and meteorological parameters were recorded at selected sites (for Hyytiälä no $SO_2$). For the ambient samples from Switzerland, OA source
apportionment contributions (presented in Daellenbach et al., 2017) were determined using an offline application of the HR-ToF-AMS, according to the protocol presented in Daellenbach et al. (2016). During the laboratory wood burning experiments, the aerosol was monitored online by an HR-ToF AMS (Aerodyne, Canagaratna et al., 2007; Bruns et al., 2016).

## 2.4 Hierarchical clustering analysis

Similarities in the mass spectral signatures were examined by hierarchical cluster analysis (HCA, Bar-Joseph et al., 2001) using the average linkage method. For a robust interpretation, we performed this analysis following two approaches, once



considering the intensity of the peaks (approach A) and once based on the presence/absence of a peak (approach B). In approach A, we computed the Pearson correlation coefficient ($R$) between the mass spectral profiles of the samples in question and performed HCA based on the correlation matrix. In approach B, we computed the number of peaks that a sample $i$ had in common with another sample normalized to the total number of peaks detected in sample $i$.

## 5  3 Results

### 3.1 Previous knowledge on aerosol composition and sources

The bulk aerosol composition for the different sites and periods is summarized in Tab. 1. On average, OC and EC loadings in Zurich were similar for the winter and summer samples. During the wood burning pollution episode the concentrations were strongly elevated at the Alpine valley sites Magadino and S. Vittore compared to the ones observed in Zurich. $NH_4^+$, $NO_3^-$,
and $SO_4^{2-}$ concentrations exhibited a strong seasonality in Zurich, with higher concentrations during winter. The same is true for NOx and $SO_2$, while higher $O_3$ concentrations were observed during summer. While $O_3$ concentrations at the remote site in Hyytiälä are comparable to Zurich, NOx and CO are strongly enhanced in Zurich highlighting the anthropogenic influence at this location. The temperatures recorded in Zurich summer (17°C) and winter (2°C) were not significantly different than those recorded during summer in Hyytiälä (between 15 and 20°C for the measurement periods) and in Magadino (1.5°C)
during wood burning episodes, respectively.

In earlier work, source apportionment analysis for the same samples (Daellenbach et al., 2017) quantified the contributions of POA from traffic (hydrocarbon-like OA, HOA), cooking (COA), biomass burning from residential heating (BBOA) as well as SOA (results for samples used in this study presented in Fig. 1). These results showed that SOA was a main contributor to
OA in the PM$_{10}$ fraction in Zurich throughout the year. We further distinguished SOA in two seasonal components termed winter OOA, WOOA, and summer OOA, SOOA, which dominate SOA in winter and summer respectively. WOOA correlated with anthropogenically-influenced inorganic ions like $NH_4^+$ and was for this reason interpreted as being formed from anthropogenic VOC emissions. SOOA in contrast showed a positive non-linear relation to temperature, consistent with the temperature driven enhancement of biogenic terpene emissions. Therefore, we have hypothesized that summer SOA is formed
from biogenic VOC emissions (Daellenbach et al., 2017, Vlachou et al., 2018), consistent with previous $^{14}$C measurements at the same and other sites in Switzerland (Zotter et al., 2014a; Vlachou et al., 2018).

### 3.2 Seasonal differences in the OA chemical composition in Zurich

The average summer ($T$>11°C) and winter ($T$<6°C) spectra from (-)ESI-UHR-MS at the urban background site in Zurich exhibited a strong seasonal difference (mass spectral signature and van Krevelen diagrams in Fig. 2). During summer, peaks
related to compounds only containing carbon, hydrogen and oxygen (CHO) dominated the spectrum. The majority of these



compounds had a ratio H/C around 1.5 and O/C between 0.4 and 1.4 (Fig. 2). These compounds were either absent or had a much lower intensity during winter.

During winter, compounds also containing nitrogen (CHON) dominated the signal (Fig. 2). The largest intensity was assigned
to $C_6H_5O_4N$ (possibly nitrocatechol), followed by $C_7H_7O_4N$ (possibly methylnitrocatechol). Some of these compounds were also present in summer with much lower contributions. Such compounds are formed through the oxidation of aromatic VOCs in the presence of NOx (Forstner et al., 1997; Jang and Kamens, 2001; Hamilton et al., 2005; Sato et al., 2012; Irei et al., 2015). In earlier work, these compounds were observed at urban and rural locations and were mainly associated with biomass burning activities (Kitanovski et al., 2012; Iinuma et al., 2010; Claeys et al., 2012; Kourtchev et al., 2013; 2014b; Zhang et al.,
2013; Mohr et al., 2013). The same peaks ($C_5H_6O_4N$ and $C_7H_7O_4N$) were also abundant in measurements in the Pearl River Delta during harvesting period (Lin et al., 2012). We did not observe nitrophenols ($C_6H_5NO_3$, $C_7H_7NO_3$, $C_8H_9NO_3$) which were previously reported for measurements with the (-)ESI-Orbitrap in a road tunnel and related to vehicular emissions (Tong et al., 2016). However, nitrophenols were also reported to be influenced by biomass burning (Mohr et al., 2013). Some aromatic CHO compounds (generic formula $C_{11}H_9O_5$) found during winter were either not detected or were only present at much lower
concentrations during summer.

Both during summer and winter, CHOS and CHONS compounds characterized by elevated H/C between 1.5 and 2 were observed, with little seasonal variability in $(S/C)_{bulk}$ (0.03). A prominent CHONS compound observed both during summer and winter was $C_{10}H_{17}O_7NS$. In early studies, this peak was linked to biogenic emissions and recent publications observed this
peak in bush fire plumes in Australia, and linked it to outgassing of BVOC during combustion (Iinuma et al., 2015). Additionally, this compound showed a high abundance during the harvest season in the Pearl River Delta (Lin et al., 2012) underlining possible additional links to human activities.

### 3.3 Comparison between Zurich and source samples

Hierarchical cluster analysis using $R^2$ (approach A) as a measure of similarity of the recorded (-)ESI-UHR mass spectral
profiles could distinguish the samples from summer and winter into two groups. On the one hand, the analysis revealed that the winter-time Zurich mass spectral signatures were similar to those recorded in Magadino and S. Vittore during wood burning episodes as well as to laboratory wood burning emissions (Fig. 3a). The spectra in S. Vittore and Magadino were most similar to the fresh laboratory wood burning emissions, suggesting the prevalence of primary wood burning emissions during these pollution episodes. The wintertime Zurich spectrum was more similar to aged laboratory wood burning emissions in the
chamber under atmospherically relevant conditions (representing 10 and 30 h atmospheric aging). On the other hand, the summertime Zurich spectrum was most similar to that from Hyytiälä during summer (Kourtchev et al., 2014a; 2016). Since Hyytiälä is strongly influenced by biogenic SOA (e. g., Kourtchev et al., 2014a; 2016), the similarity between the mass spectral signatures for summertime Hyytiälä and summertime Zurich suggests that biogenic SOA has also a dominant influence at the



urban background location in Zurich. However, the correlation between summertime Zurich and Hyytiälä was smaller than the ones between wintertime Zurich, Magadino and S. Vittore, indicating differences either in the SOA precursor emission patterns or in the SOA formation pathways.

5 As for any chemical ionization mass spectrometry, the spectral fingerprints are influenced by the variable relative ionization efficiencies of the different compounds. Therefore, we also performed hierarchical cluster analysis based on the normalized number of common peaks (approach B, Fig. 3b). Results confirmed the similarity of samples dominated by wood burning emissions (laboratory wood burning experiments, S. Vittore and Magadino), clearly distinguished from other samples. Among the wood burning dominated samples, fresh laboratory wood burning emissions were most similar to S. Vittore and to a lesser 10 degree to Magadino. In addition, the analysis indicates a high number of peaks with relatively low intensity in common to the samples from Zurich during summer and winter. As will be discussed in more detail in the following sections these peaks comprise CHONS compounds. Overall, the appearance of peaks related with biogenic SOA and wood burning emissions in summer and winter, respectively dominate the observed variability between the spectra. Therefore, samples from summer and winter will be discussed separately in the following.

## 3.4 Biogenic SOA: Zurich and Hyytiälä

### 3.4.1 Bulk chemical composition

In order to understand the spectral profiles recorded in summertime Zurich, we start by describing similar spectra from Hyytiälä. The largest fraction of signal in Hyytiälä can be attributed to CHO compounds with H/C ~1.5 and $C_{bulk}$ 10.5 (H/C 1.48, $C_{bulk}$ 9.7 for 2011 and H/C 1.50, $C_{bulk}$ 11.3 for 2014, Fig 4, 5, Tab. 2), characteristic of biogenic emissions. Compounds 20 with 8 to 12 carbons (C8-C12), thought to arise from monoterpene oxidation, dominate the signals (Fig. 5, 6, 7). Additionally, CHO compounds with 13-16 carbons and 17-22 carbons significantly contribute to the signal. The C13-C16 compounds are thought to consist mainly of sesquiterpene oxidation products, but may also be produced through accretion reactions of monoterpene and isoprene $RO_2$ radicals (Berndt et al., 2018). Meanwhile, the latter class (C17-C22) of compounds is thought to consist mainly of dimeric oxidation products (e.g. Kristensen et al. 2016; Berndt et al., 2018; Frege et al., 2018). Larger 25 oxidation products (C>22), were detected but to a much lower extent.

The OA bulk composition in summertime Zurich was similar to that in Hyytiälä, especially in the C8-C12 range, and at both locations CHO compounds dominated the signal (Fig. 4, 5, 6). The $(H/C)_{bulk}$ was 1.49 for Zurich samples during summertime, which is similar to those for samples from Hyytiälä and to the oxidation products of biogenic VOCs (e.g. α-pinene: pinic acid: 30 $C_9H_{14}O_4$, 3-methyl-1,2,3-butanetricarboxylic acid (MBTCA, $C_8H_{12}O_6$). However, compared to Hyytiälä, signals in Zurich were clearly shifted toward smaller molecules ($C_{bulk}$ 7.6, Fig. 4, 6, 7, Tab. 2). While a high fractional contribution to the signal of C13-C16 and C17-C22 compounds was observed in Hyytiälä 2014, these compounds contributed much less in Zurich (Fig.



5, 6, 7). Meanwhile, small molecules such as $C_4H_6O_5$ (possibly related to malonic acid) and $C_5H_8O_5$ (possibly related to hydroxyglutaric acid) exhibited a higher fractional contribution in Zurich during summer than in Hyytiälä 2014 (Fig. 4, 6, 7). Some of these compounds were related to OH radical induced atmospheric aging of monoterpene SOA, especially at high NOx conditions, in ambient as well as in laboratory experiments (Zhang et al., 2018; Mutzel et al., 2015). In the following, we will

discuss the possible reasons for the differences.

### 3.4.2 CHO and BVOC composition

The composition of BVOC emissions depends on various parameters such as vegetation type and temperature. While many BVOCs lead to the formation of oxidation products characterized by H/C ~1.5, $C_{bulk}$ depends on the size of the carbon backbone of the initially emitted precursor and the degree of accretion. Thus, the composition of the BVOC emissions has an impact on

$C_{bulk}$. Modelled biogenic emissions showed a higher isoprene (ISO, $C_5H_8$) to monoterpene (MT, $C_{10}H_{16}$) ratio in Switzerland than in Finland (Fig. S2, Jiang et al., in prep.). The higher ISO/MT ratio in BVOC emissions in Zurich could contribute to the higher C3-C7 CHO compound contribution at this site (see above, Fig. 4, 5, 7). SQT/MT did not show a clear difference between Finland and Switzerland and is therefore not expected to be the reason for the observed enhanced abundance of C13-C17 compounds in Hyytiälä compared to summertime Zurich (however, see the NOx discussion in the next section). In

addition, increased temperatures lead to higher BVOC emissions, and may induce some effects on the ratios of ISO/MT (mainly driven by photosynthetic activity) and of SQT/MT (Zhao et al., 2017). Biotic stress acting on plants may influence the SQT/MT emission ratio as well: Zhao et al. (2017) reported ratios of 0.15 for unstressed and 3.5 for stressed plants at 22-25°C. These various effects could contribute to the variability in the abundance of C13-C17 CHO compounds observed between Zurich (17°C, average $T_{max}$=21°C) and Hyytiälä in 2011 (15°C, average $T_{max}$=18°C) and 2014 (20°C, average $T_{max}$=24°C).

### 3.4.3 CHO and temperature

In 2014, higher daily average temperatures (20°C, average $T_{max}$=24°C) were recorded compared to 2011 (15°C, average $T_{max}$=18°C). The mass spectral signature recorded in Hyytiälä in 2011 showed distinct differences to the observations made in 2014 at the same place and was also less similar to summertime Zurich (Fig. 7a). In 2011 (lower T), C8-C12 compounds dominated the signal while larger compounds (C13-C16 and C17-C22) contributed significantly to the signal in 2014 (Fig. 5).

As highlighted by Kourtchev et al. (2016) the higher contributions of dimeric and trimeric BVOC oxidation products in 2014 could be related to higher precursor and SOA mass which is in agreement with laboratory experiments presented in the same study. Temperature differences affect not only the emissions from the biosphere, but also the ratio between particle and gas-phase concentration of compounds as a function of their volatility. This could lead to an enhancement of less volatile dimeric compared to more volatile monomeric BVOC oxidation products at higher T (Fig. 7b, Hyytiälä 2014 vs 2011). Temperature

affects also the particle phase contribution of 1st (more volatile) and 2nd (less volatile) generation gas-phase oxidation products (Zhang et al., 2010; Vogel et al., 2013; Müller et al., 2012; Donahue et al., 2012). This is consistent with the observed





enhancement of MBTCA compared to pinic acid with rising temperatures is consistent (Fig. 7c). This phenomenon partially explains the variability in the observed composition of monomeric BVOC oxidation products.

Kourtchev et al. (2016) observed an increasing fraction of smaller molecules (C3-C7) at higher temperatures. The increase in the proportion of smaller compounds (C3-C7) occurs despite their increasingly higher evaporation rates. This could be related to a higher fraction of 1$^{st}$ generation products residing in the gas-phase where they are prone to further oxidation, possibly also promoting fragmentation. Since the average temperature in Zurich during summer is 17°C (average $T_{max}$=21°C) this would partially explain the enhancement of the fraction of lower molecular weight compounds (C3-C7) compared to Hyytiälä.

### 3.4.4 CHO and NOx

While laboratory monoterpene experiments show an important influence of functionalized monomeric oxidation products, ambient measurements have revealed an enhancement of fragmentation over functionalized products with increasing NOx concentrations (Zhang et al., 2018). Fragmentation products of $RO_2$ + NO reactions and subsequent autooxidation could explain such observation. Since we observe a similar behavior (Fig. 7b) in this study, the higher (C3-C7)/(C8-C12) ratio in summertime Zurich than in Hyytiälä can be related to enhanced NOx concentrations at the urban site (NOx summertime Zurich: 15 ppb, Hyytiälä: 0.5 ppb).

Dimeric monoterpene oxidation products (C17-C22) are mainly formed through $RO_2$ + $RO_2$ reactions, while in the presence of NOx this reaction pathway is suppressed by radical termination reactions between $RO_2$ and NO (Kristensen et al., 2016, Lehtipalo et al., 2018). This effect explains the considerable contribution of C17-C22 compounds in Hyytiälä, while they are largely absent in summertime Zurich (Fig. 5, 7). Overall the enhanced NOx concentrations inhibit the formation of such dimeric C17-C22 compounds leading to the smaller $C_{bulk}$ in summertime Zurich than in Hyytiälä.

### 3.4.5 CHON, CHOS, CHONS

A fraction of the signal is related to compounds also consisting of nitrogen and/or sulfur (CHON, CHOS, and CHON) in summertime Zurich (32%) and Hyytiälä (19, 2011: 15%, 2014: 23%). Both in in Hyytiälä and summertime Zurich CHOS compounds contribute to the signal (Zurich: 14%, Hyytiälä 2011: 6%, 2014: 22%), while the summertime Zurich $SO_2$ concentration (average 0.4 ppb) exceeds a typical Hyytiälä concentration (June-July-August Q75=0.15 ppb, during sampling periods not available). Such compounds were also detected in wintertime Zurich. This group of compounds exhibits a similar composition at both sites (H/C range 1.3-1.9, C5-C8 and C9-C12, though also some larger compounds were found in Hyytiälä 2014 as expected with the enhanced contribution of dimers). CHON compounds are enhanced in summertime Zurich (CHON: 13%) compared to Hyytiälä (CHON: 4%, 2011: 7%, 2014: 1%) but contribute clearly less than in wintertime Zurich. The CHON compounds have a similar H/C (0.8-0.9) at both sites but a higher $C_{bulk}$ in Hyytiälä (C9-C12) than in summertime Zurich (C5-C8). CHONS compounds are observed in summertime Zurich (4%) and cover an H/C range similar to CHOS



compounds (1.4-1.9) but the signal can almost uniquely be explained by compounds with C9-C12 (most prominently $C_{10}H_{17}O_7NS$). These compounds are largely absent in Hyytiälä (1%) which might be explained by elevated NOx concentrations in Zurich.

**3.5 Wood burning emissions: laboratory experiments, ambient pollution episodes, and winter-time pollution**

**3.5.1 Chemical composition**

Wood burning is an important wintertime source of OA in Central Europe (Herich et al., 2014; Lanz et al., 2010; Crippa et al., 2014; Zotter et al., 2014a; Daellenbach et al., 2017). In the following, filters from laboratory wood burning experiments were used as a reference for understanding the influence of such emissions on wintertime pollution at different sites (Fig. 8, 9). During laboratory wood burning experiments, aromatic CHON compounds with H/C between 0.8 and 1.0 and 5 to 8 carbons
(C5-C8) contribute a large fraction of the signal (43%, Fig 4, 5). Additionally, CHO compounds, not present in summer samples, with C8-C12 and H/C 0.8-1.0 contribute significantly to fresh emissions (56%). Since the contribution of CHON to biomass burning aerosol increases at lest initially during aging (65-78% for aged and 43% for fresh emissions), the relative contribution of CHO decreases (22-34% for aged and 56% for fresh emissions) (Fig. 5). The composition during wood burning episodes in Alpine valleys is similar to primary wood burning emissions sampled in the laboratory (CHON: 47-58%, CHO:
48-41%, Fig. 5). In wintertime Zurich, the chemical composition is also characterized by a large contribution of CHON (43%) and CHO (35%) compounds to the signal but smaller than at the Alpine valley sites (Fig. 5). However, additionally also CHOS (12%) and CHONS (10%) compounds contribute to the signal in wintertime Zurich, which is neither the case in the laboratory wood burning experiments nor wood burning episodes in the Alpine valleys (Fig. 5, 9). The composition of the laboratory wood burning emissions, wood burning episodes in Alpine valleys and wintertime Zurich are clearly distinguishable from
biogenic SOA by a higher contribution of CHON compounds (43-58%) as well as a lower bulk H/C (0.93-1.24) and $C_{bulk}$ (7.2-8.7) – than for biogenic SOA (CHON: 1-4%, H/C: 1.48-1.50, $C_{bulk}$: 9.7-11.3) (Fig. 4, 5).

During the laboratory wood burning experiments, compounds with H/C of 0.8-1.0 and 5-8 carbons dominate the CHON family. In fact, only a few compounds such as nitrocatechols and similar compounds ($C_6H_5O_4N$, $C_7H_7O_4N$, $C_8H_9O_4N$) contribute to
the signal. Compounds detected in a road traffic tunnel which were related to vehicular emissions ($C_7H_7O_3N$ and $C_8H_9O_3N$, Tong et al., 2016) were also detected in the primary wood burning emissions but with much lower intensities than $C_7H_7O_4N$ and $C_8H_9O_4N$. While SOA formation from vehicular emissions might proceed via the oxidation of aromatic species, the most important precursors in biomass smoke are oxygenated aromatics such as phenol, cresol, and catechol (Harrison et al., 2005; Platt et al., 2013; Bruns, et al., 2015; 2016; 2017, Schauer et al., 2001 and 2002). While CHON compounds were already
present in the fresh emissions and thus were directly emitted, the increasing $(N/C)_{bulk}$ indicates a strong additional secondary formation of such compounds (Fig. 4, 5, 8). In Magadino, S. Vittore and wintertime Zurich similar CHON compounds dominated the signal. Their composition suggests that the biomass burning emissions observed in wintertime Zurich were



further processed than in Magadino and S. Vittore. During summer these compounds exhibited a much smaller contribution in Zurich (see more detail in Section 3.4, and Fig. 5).

The CHO compounds observed in the fresh wood burning emissions during the laboratory experiments were characterized by a lower H/C (0.7 to 1.0) than biogenic SOA (1.5) (Fig. 5, 9). Anhydrous sugars with high H/C such as levoglucosan ($C_6H_{10}O_5$, H/C 1.67, O/C 0.83), mannosan and galactosan directly emitted from cellulose pyrolysis during biomass burning were detected in the laboratory wood burning emissions as well as in Magadino and S. Vittore but contributed only little to the signal. A considerable amount of the CHO compounds in wood burning emissions could be considered aromatic ($X_c \geq 2.5$) or even condensed aromatic ($X_c \geq 2.7143$, Fig. S4). These compounds are consistent with products from lignin pyrolysis (Fig. S4, Bertrand et al., 2017, 2018) and contributed significantly to the CHO signal recorded for primary wood burning emissions but less for aged emissions (57%, 49%, 44% of CHO aromatic, 5%, 3%, 3% condensed aromatic, 38%, 48%, 53% non-aromatic for fresh, 10 h, and 30 h atmospherically aged emissions, respectively). The ambient wood burning pollution showed a similar distribution of the CHO signal as aged laboratory wood burning emissions (Magadino: 46% aromatic, 6% condensed aromatic, 48% non-aromatic, S. Vittore: 44% aromatic, 4% condensed aromatic, 52% non-aromatic) and the detailed chemical composition was similar to the fresh laboratory wood burning emissions (Fig. S4). With proceeding aging during the wood combustion experiments, more oxygenated compounds dominated the signal during the wood combustion experiments (center at H/C of 1.0 and O/C of 0.5, Fig. 9, S4). These values are consistent with aqueous SOA from syringol/guaiacol/phenol formed through the reaction with hydroxyl radicals and excited states of organic compounds (Yu et al., 2014). Since wood burning is a known emitter of such compounds, it seems probable that the aged wood burning emissions consisted of oxidation products of phenolic compounds (Schauer et al., 2001; Bruns et al., 2016; 2017). In wintertime Zurich, the CHO composition shows largely common features with aged laboratory wood burning emissions (Fig. 9, S4). However, in wintertime Zurich the contribution of smaller CHO compounds (C3-C7) was higher (Fig. 5, 7, S3) and the contribution of aromatic compounds to the total CHO signal was clearly lower (22% aromatic, 2% condensed aromatic, 76% non-aromatic) than for the laboratory wood burning emissions (Fig. S3, S4, S5, S6). This indicates that additional processes leading to fragmentation play a role in the urban environment as already observed in summer. During summer the influence of aromatic compounds on the CHO signal was negligible (Zurich summer: cond. arom: 0%, arom: 3%, non-aromatic: 97%).

In wintertime Zurich, CHOS and CHONS compounds contributed significantly to the signal as opposed to the laboratory wood burning experiments and Alpine valleys. These compounds had between 9 and 12 carbons and were characterized by an H/C between 1.5 and 2.0. During the laboratory wood burning experiments (aging initialized with HONO) no $SO_2$ was added during photochemical aging which could explain the absence of these CHOS and CHONS compounds. On the other hand, a compound group with similar H/C and $C_{bulk}$ was also found in summertime Zurich but not in biogenic SOA. Thus they can neither be linked to biogenic SOA nor wood burning emissions. The presence of this compound group indicates the importance of additional sources and/or processes in the formation of urban SOA.



### 3.5.2 Atmospheric aging of wood burning OA

Nitrocatechol ($C_6H_5O_4N$) and methylnitrocatechol ($C_7H_7O_4N$) are two molecules commonly studied in ambient aerosol samples and used as markers for wood burning SOA (Iinuma et al., 2010). These compounds were large contributors to the signal during the laboratory wood burning experiments as well as in wintertime Zurich, Magadino and S. Vittore (Fig. 5, 8).

Additionally, in the laboratory experiments the signal ratio of $C_6H_5O_4N$ to $C_7H_7O_4N$ increased with atmospheric aging (Fig. 8, H/C 0.8 and 1.0). Based on this observation, we conclude that the wood burning emissions analyzed in S. Vittore and Magadino are fresher than the ones in wintertime Zurich. In the following, we directly link the signal ratio $C_6H_5O_4N/C_7H_7O_4N$ to the atmospheric age of wood burning emissions (Fig. 10). We derived a relation between the aging time in the smog chamber (10 and 30 h equivalent atmospheric aging) and the $C_6H_5O_4N/C_7H_7O_4N$ intensity ratio. Thereby, we estimated that in winter

Zurich the wood burning emissions were on average aged for 9.8 h (range 5.9-13.1h). In contrast, at the Alpine valley sites the wood burning emissions were only aged between 0.3 (S. Vittore, range 0-4.4h) and 1.0 h (Magadino, range 0-4.8h). In a next step, we approximated the contribution of POA to OA during the smog chamber experiment using the OA (AMS) and BC (aethalometer) measurements and assuming a constant POA/BC ratio during the measurement time yielding a POA/OA of 0.22 after 10h atmospheric aging and of 0.14 after 30h. With this parametrization and the approximated aging time of the wood

burning emissions for the ambient samples we estimated the contribution of primary wood burning OA (wbPOA) to the total wood burning OA (wbOA) for the ambient data (Fig. 10). The POA contribution to wbOA was 22% in Zurich winter (best estimate by linear regression, range 18-36% based on uncertainty of estimated atmospheric aging time) but much higher in Magadino 90%, range 44-100%) and S. Vittore (100%, range 47-100%). Based on the offline AMS PMF, we use BBOA as an estimate of wbPOA and WOOA as an upper limit of wbSOA. At the Alpine valley sites, the influence of BBOA to the sum

of BBOA and WOOA was higher than in winter-time Zurich (S. Vittore: 100% BBOA, Magadino. 90% BBOA, and winter-time Zurich 27% BBOA). These results are consistent with the findings in this study.

### 3.6 Temporal behavior

The relative contribution of compounds with H/C between 1.2 and 1.7 (characteristic range of BVOC SOA, Fig. 11) explained largely the signal in Hyytiälä (approximately 75%) and contributed only little to the winter-time Alpine valley samples and

laboratory wood burning experiments (bulk composition in Fig. S7). The relative contribution of this compound class showed a seasonal behavior similar to the local temperature in Zurich ($R^2$=0.75, p<10⁻⁴). The higher the temperature the larger was the contribution of small compounds (C3-C7). Additionally, also some CHON, CHOS and CHONS molecules were part of this compound class which suggests a biogenic origin of these compounds. In Zurich, a good correlation was observed between the relative signal contribution of the compounds with H/C between 1.2 and 1.7 and the relative contribution of SOOA to OA

($R^2$=0.57, p<0.01).



On the other hand, a large part of the signal of the winter-time Alpine valley samples and the laboratory wood burning samples could be explained by a compound class characterized with H/C between 0.7 and 1.1 (characteristic range of wood burning emissions, Fig. 11). The largest contribution to the relative signal of this group was from CHON molecules. In Zurich, the relative contribution of this compound class to the signal had a seasonal pattern consistent with the residential heating behavior.

In Zurich, a good correlation was observed between the relative signal contribution of the compounds with H/C between 0.7 and 1.1 and the relative contribution of the sum of BBOA and WOOA to OA ($R^2$=0.59, p<0.001), with lower correlations with either BBOA alone ($R^2$=0.29, p=0.02) or WOOA alone ($R^2$=0.37, p<0.01).

## 4 Conclusions

This work links seasonal variability in OA composition based on negative ESI-Orbitrap data from an urban background site
in Central Europe (Zurich, Switzerland) to wood burning emissions in winter and to biogenic SOA in summer. The mass spectral signatures observed for laboratory wood burning emissions were dominated by CHON compounds ($C_6H_5O_4N$, $C_7H_7O_4N$). The influence of CHON increased during aerosol aging while the relative contribution of CHO compounds decreased as the CHO content became less aromatic (with the aromatic fraction of the CHO signal accounting to 62% in the fresh aerosol and 47% in 30h aged aerosol). Signatures from winter-time pollution episodes at two Alpine valley sites could
be explained by the laboratory experiments. Winter-time Zurich signatures were also dominated by CHON compounds but the CHO fraction showed some differences to the laboratory wood burning experiments and ambient wood burning pollution episodes (less aromatic and higher contribution of C3-C7 compounds in Zurich). Additionally, in Zurich a considerable influence of CHOS and CHONS was observed throughout the year which suggests that additional sources and/or processes are important in Zurich. The summer-time signature from Zurich was dominated by CHO compounds and showed a similar
mass spectral signature as biogenic SOA observed in the boreal forest (Hyytiälä, Finland). C3-C7 compounds contributed a larger relative fraction to the signal in Zurich than in the boreal forest. While compounds related to sesquiterpenes (C13-C16) and dimers of α-pinene oxidation products (C17-C22) were prominent in the boreal forest, they were largely absent in Zurich during summer. These observations may be explained by differences in oxidant concentrations (mostly NOx) and in the composition of biogenic VOC emissions.

*Data availability:* The data are available upon request from the corresponding author.

*Author contribution:* Conceptualization: KRD, IEH, ASHP, IK, MK formulated the study. Investigation and data curation: KRD, IK performed the Orbitrap analyses, JLJ performed quantified MBTCA and pinic acid, EAB performed the smog
chamber experiments, TP curated data related to Hyytiälä, JJ and SA performed the model calculations. Formal analysis, methodology, visualisation: KRD, IEH designed and performed the statistical analysis and data visualisations. Validation: KRD, IEH, IK, ALV. Software, methodology: KRD, IEH, IK. Writing: KRD wrote the original draft, which was reviewed,




commented and edited by all the authors. Funding acquisition, resources, project administration, supervision: ASHP, UB, MK supported and supervised the research.

*Competing interests.* The authors declare that they have no conflict of interest.

*Acknowledgements.* This work was supported by the Swiss Federal Office of Environment; Liechtenstein; Ostluft; the Swiss cantons Basel, Graubünden, Thurgau, the Competence Center Environment and Sustainability (CCES) (project OPTIWARES) and the Swiss National Science Foundation (WOOSHI grant 140590). K. R. Daellenbach acknowledges the support by SNF
grant P2EZP2_181599.

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

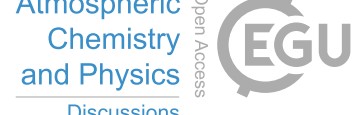



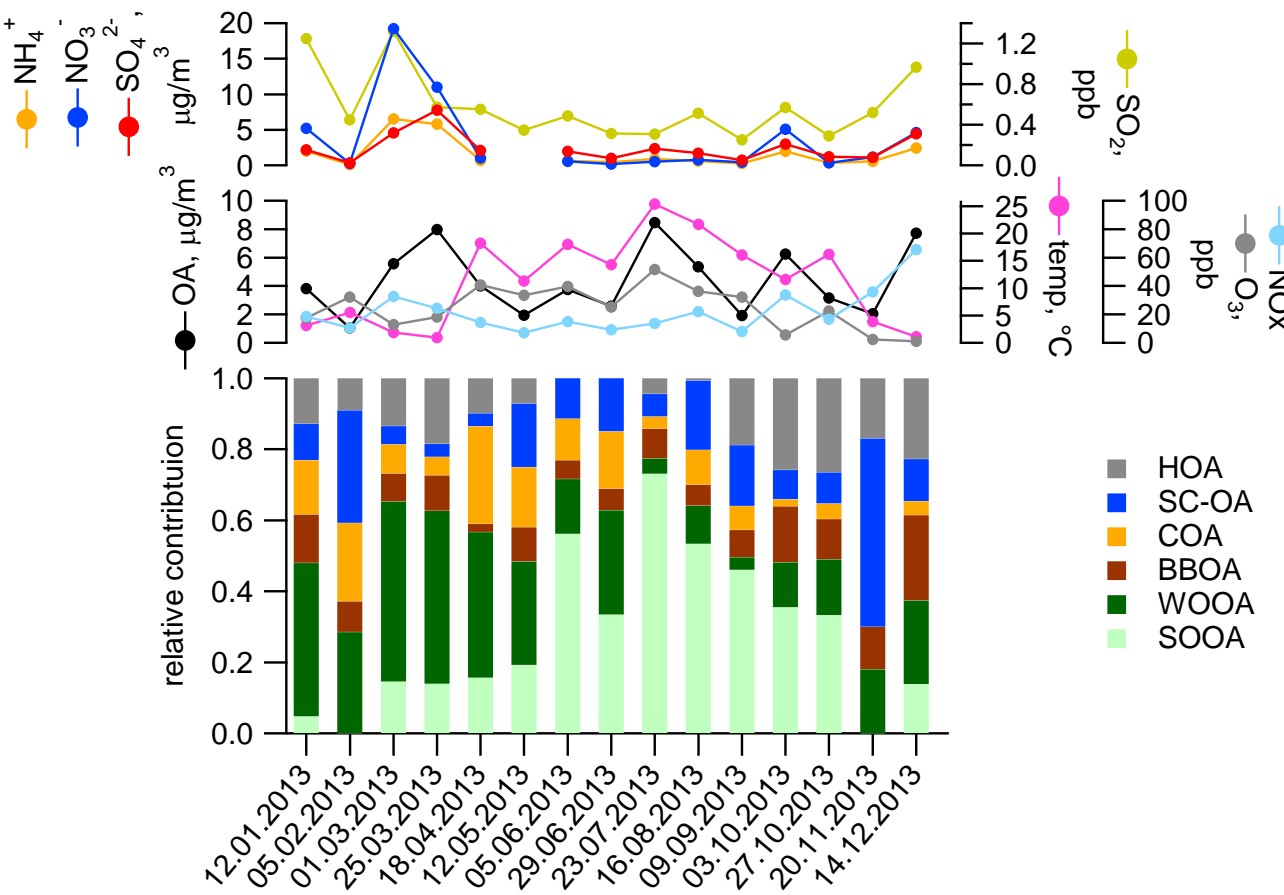

Figure 1: Concentrations of particulate and gaseous species (upper and middle panels) and relative source contributions to the organic aerosol as determined by offline AMS in Zurich (lower panel): traffic (HOA), cooking (COA), and wood burning POA (BBOA), as well as a factor explaining sulfur-containing organic fragments (SC-OA), and SOA categories dominant in summer (SOOA) and winter (WOOA).



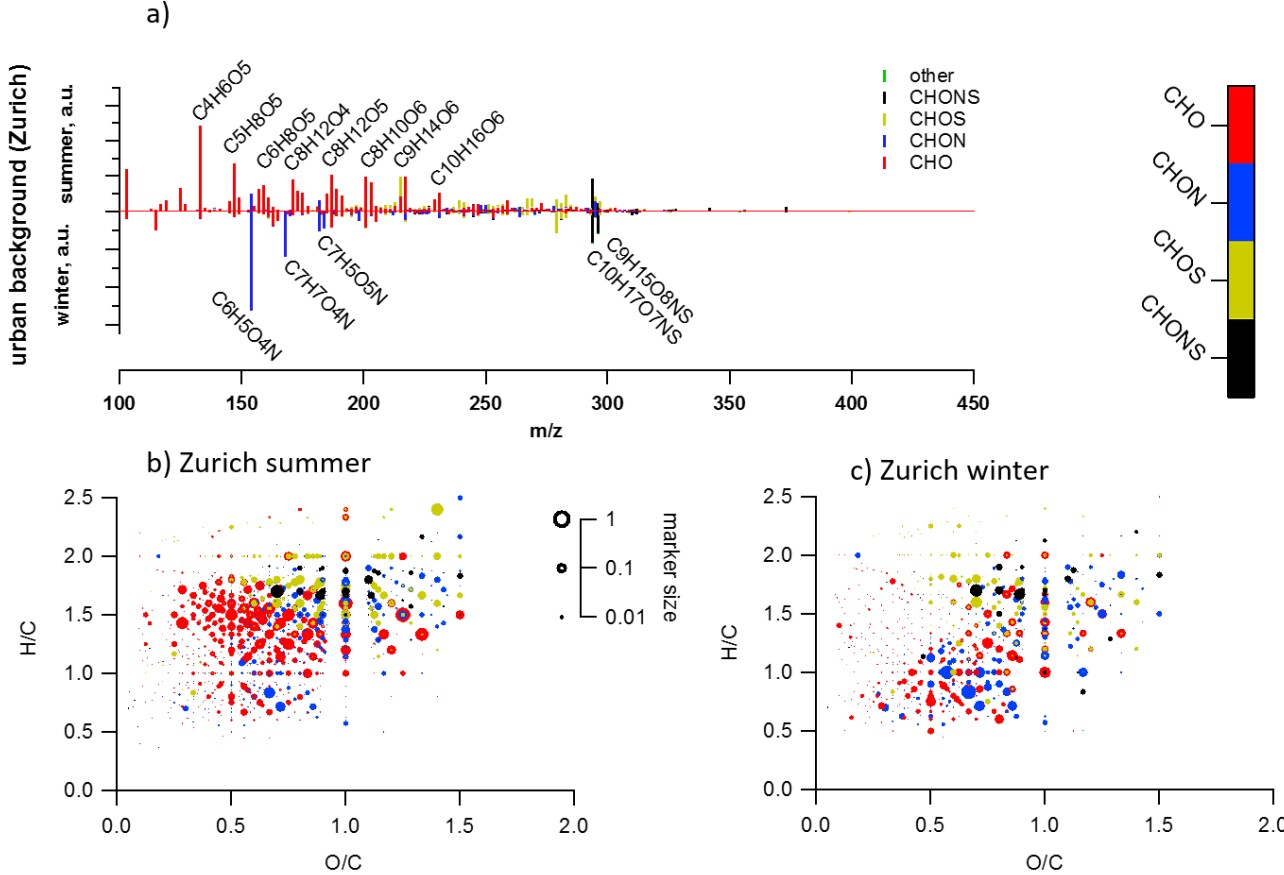

Figure 2: a) Average summer ($T$>11°C, OA mass-weighted, 9 samples) and winter ($T$<6°C, OA mass-weighted, 6 samples) ultra-high-resolution mass spectra integrated to unit-mass resolution in the negative mode for the organic aerosol in $PM_{10}$ sampled in Zurich during the year 2013 (weighed average with OA concentration from offline AMS analysis). The signal at

5    a nominal mass is separated by ion family (CHO, CHON, CHOS, CHONS, other). Peak assignments of dominant ions of selected UMR peaks are labelled as neutral compounds. In the absence of signal, the mass spectra are only plotted until $m/z$ 450 not 650. b) Van Krevelen diagrams (neutral composition) of average summer ($T$>11°C, OA mass-weighted, 9 samples) and winter ($T$<6°C, OA mass-weighted, 6 samples) ultra-high-resolution mass spectra in the negative mode for the organic aerosol in $PM_{10}$ sampled in Zurich during the year 2013 (weighed average with OA concentration from offline AMS analysis

10  ). Peaks are displayed as circles with their size reflecting log(intensity) and the color-code the molecular composition of the compound.



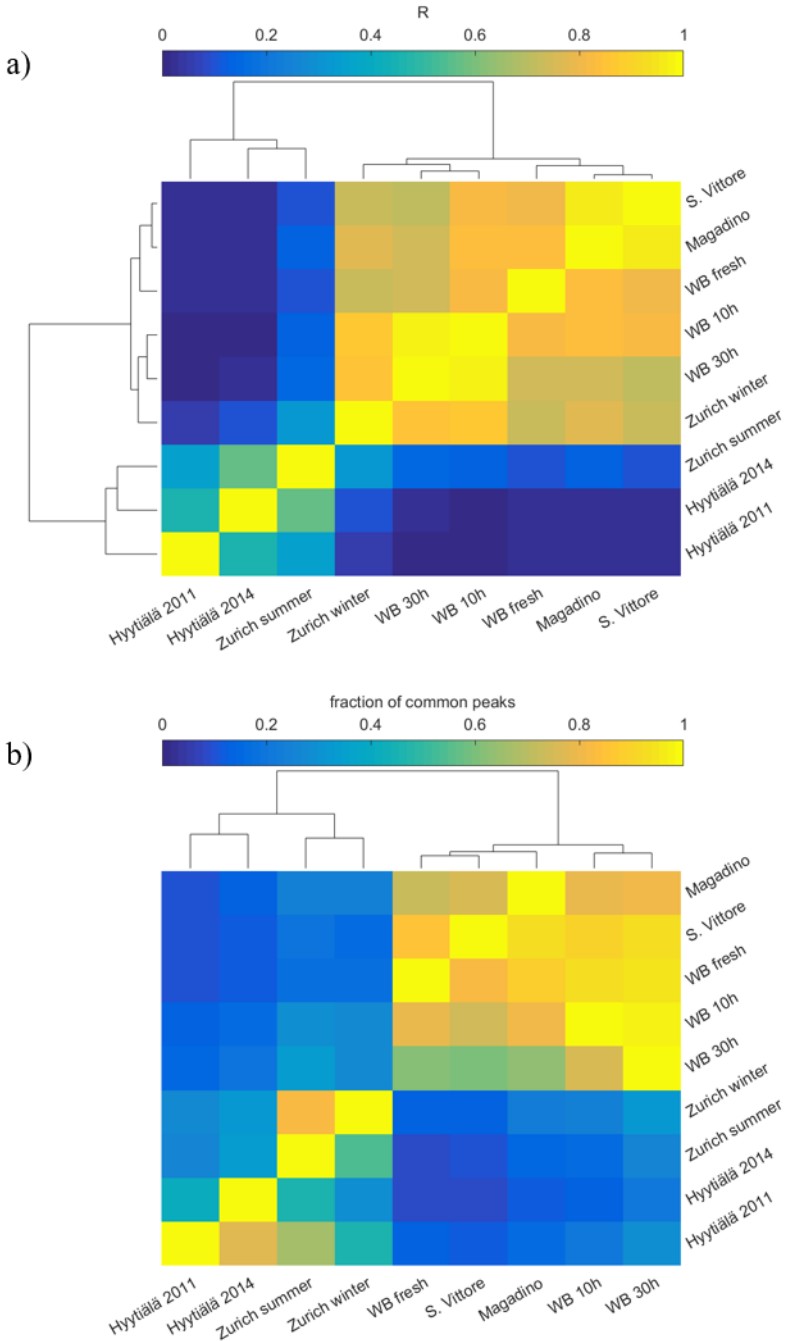

Figure 3: a) correlation matrix of mass spectra sorted by hierarchical cluster analysis also depicting the similarity as dendrograms, b) number of common peaks normalized to the total number of peaks of the respective sample sorted by

5   hierarchical cluster analysis also depicting the similarity as dendrograms.



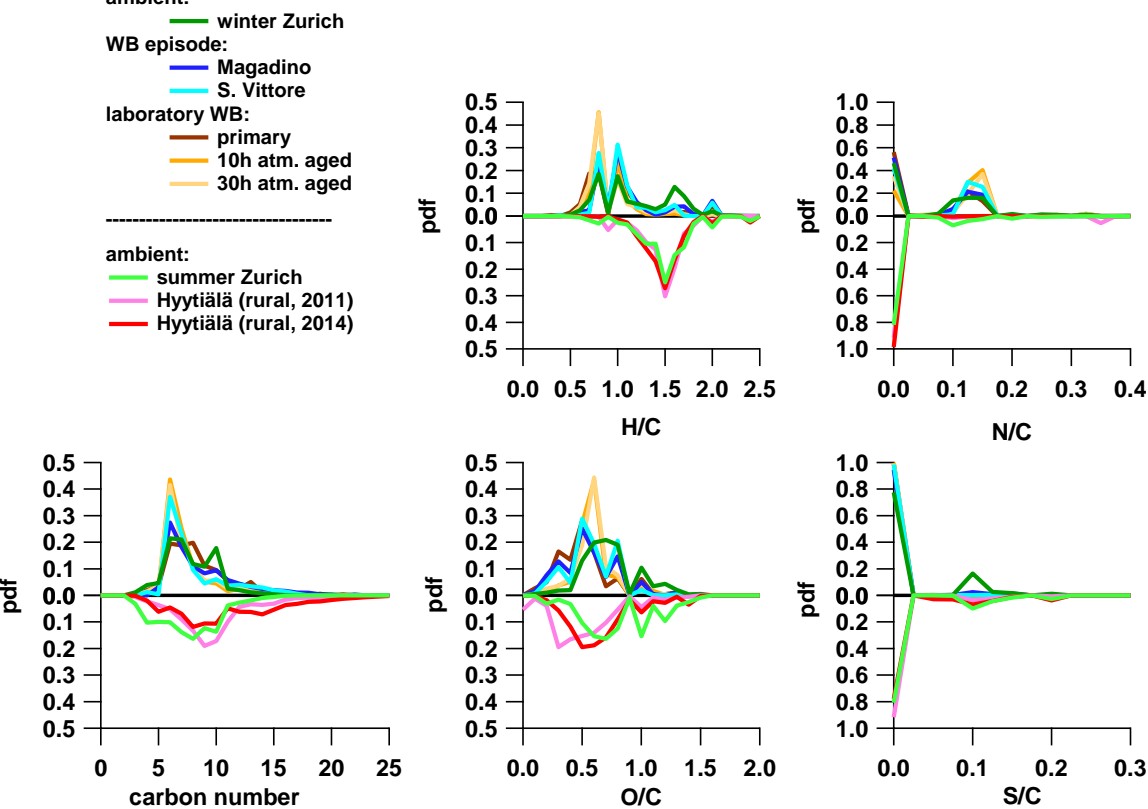

Figure 4: Probability density functions (pdf) of bulk properties of organic aerosol (neutral composition based on (-)ESI-ultra-high resolution mass spectra) for different emission and aging conditions: number of carbon, H/C, O/C, N/C, S/C of ambient samples from Zurich in winter and summer, wood burning episodes at the Alpine valley sites Magadino and S. Vittore, and laboratory wood burning experiments (fresh emissions and after simulated atmospheric aging of 10 and 30h) as well as from the boreal forest in Hyytiälä, Finland (Kourtchev et al., 2014a, 2016). The carbon number is binned in ranges of rounded masses, H/C and O/C in ranges of 0.1, N/C and S/C in ranges of 0.025





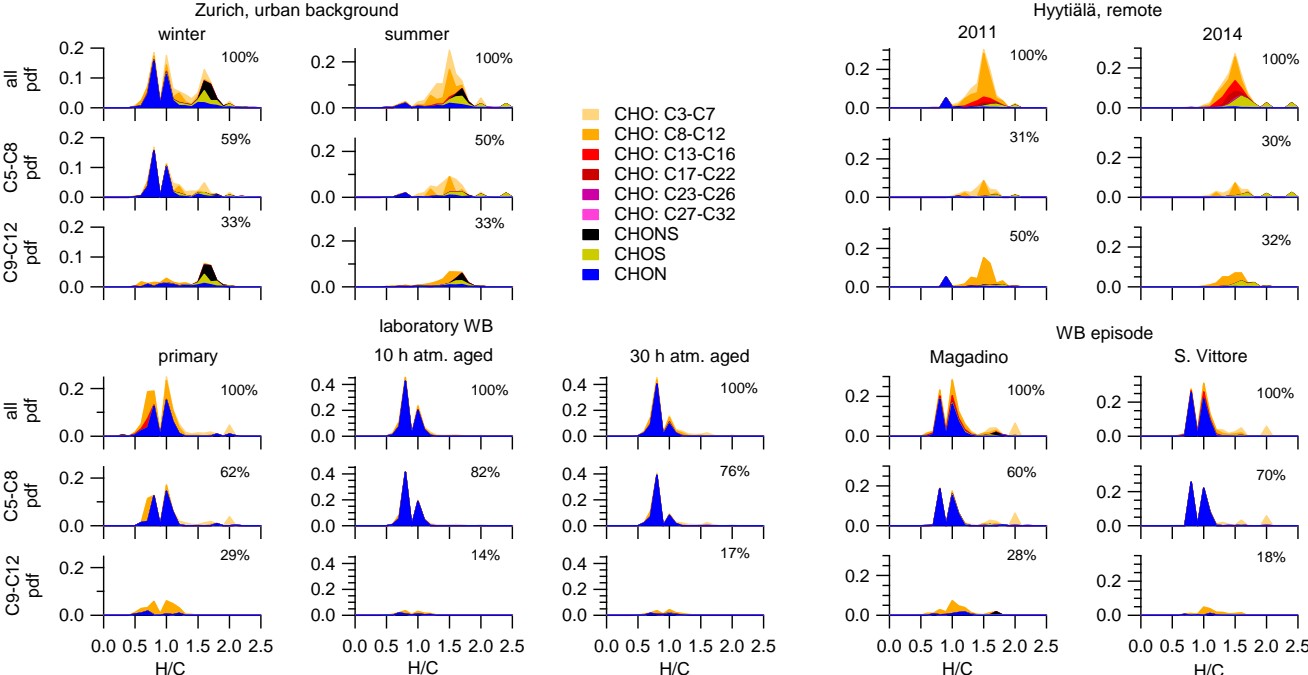

Figure 5: Probability density functions (pdf) and contributions of different molecule families to H/C for all molecules (neutral composition based on (-)ESI-ultra-high resolution mass spectra), molecules with 5 to 8 carbon atoms, and 9 to 12 carbon atoms for the ambient samples collected in Zurich, Magadino, S. Vittore, and wood burning smog chamber experiments. The area of

5   the histograms is proportional to the percentage of the total signal explained for each data set.





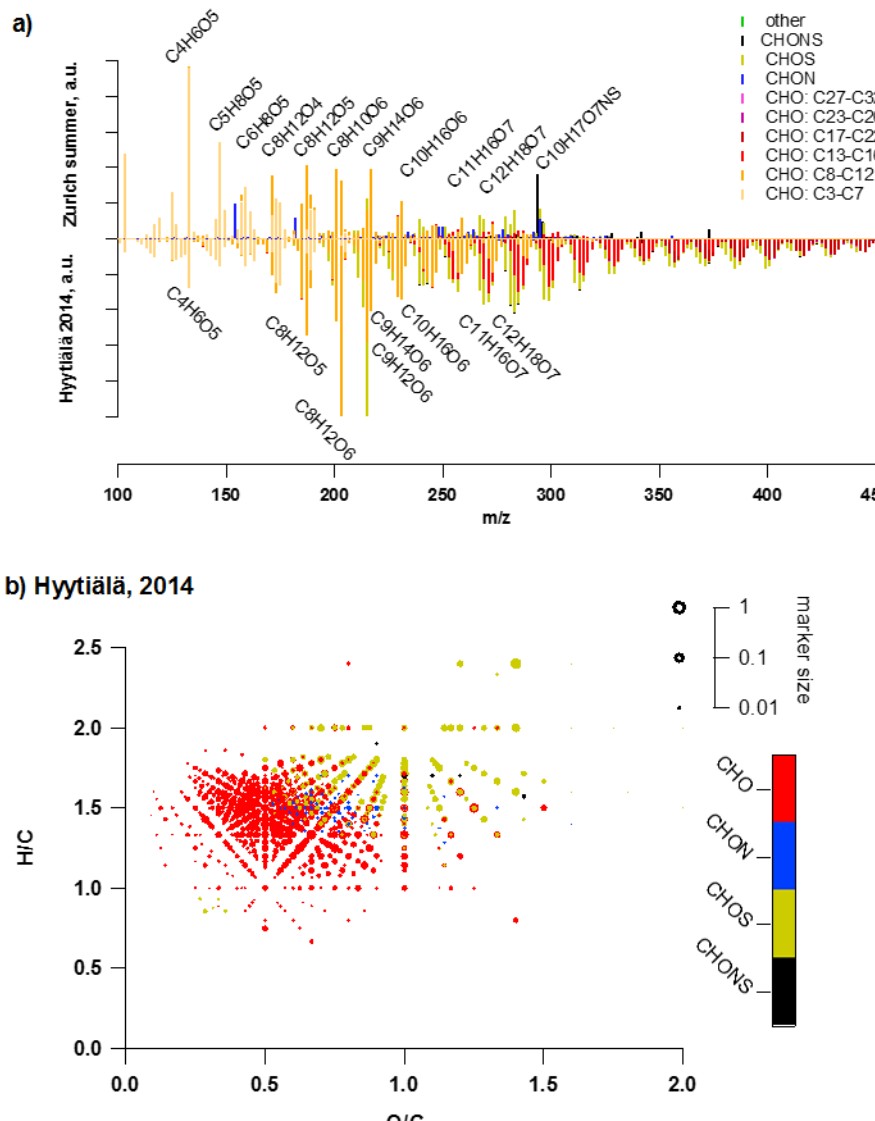

Figure 6: a) Average summer (T>11°C) ultra-high-resolution mass spectra integrated to unit-mass resolution in the negative mode for the organic aerosol in $PM_{10}$ sampled in Zurich during the year 2013 (weighed average with OA concentration from offline AMS analysis) compared to the equivalent spectrum from Hyytiälä 2014. The signal at a nominal mass is separated by ion family (CHO, CHON, CHOS, CHONS, other). Peak assignments of dominant ions of selected UMR peaks are labelled as neutral compounds. b) Van Krevelen diagram of negative ion mode spectra (neutral composition) of Hyytiälä 2014 sample peaks are displayed as circles with their size reflecting log(intensity) and the color-code the molecular composition of the compound (data from Kourtchev et al., 2016).



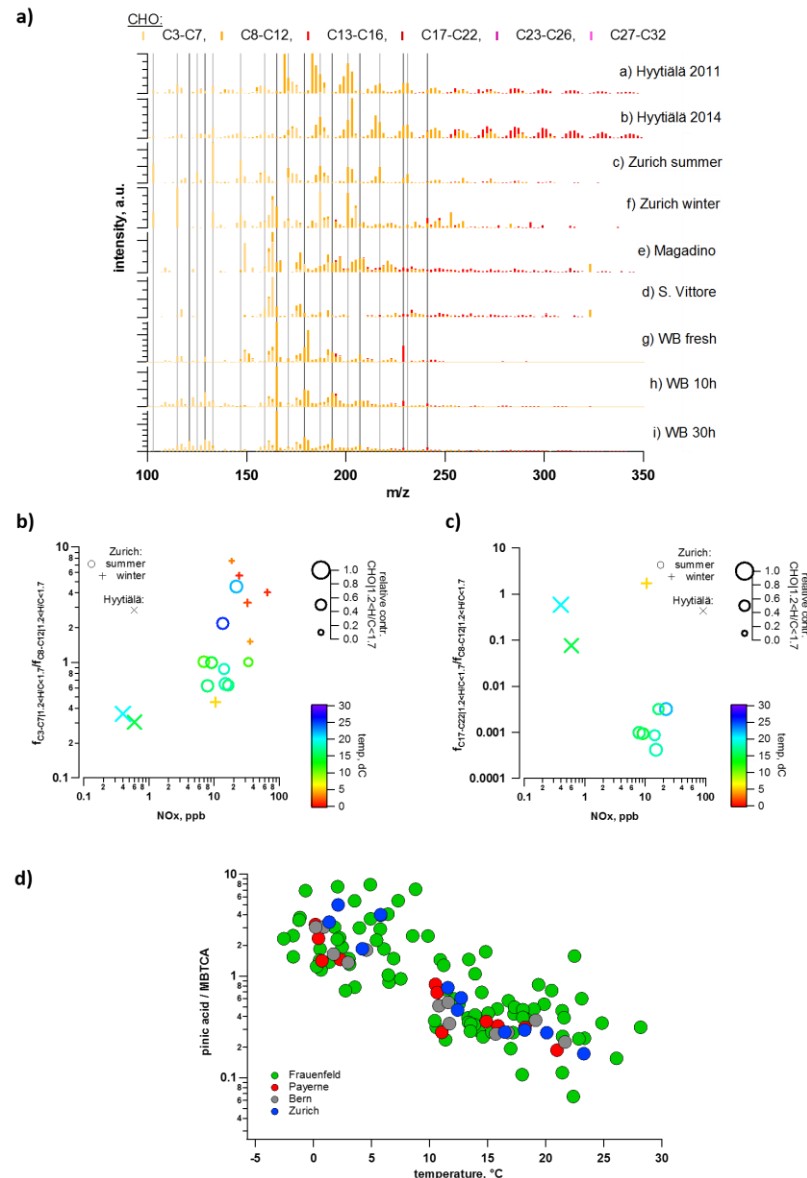

Figure 7: a) Ultra-high-resolution mass spectra of CHO compounds integrated to unit-mass resolution in the negative mode for the organic aerosol for Zurich during summer ($T$>11°C) and winter ($T$<6°C) 2013 (PM$_{10}$, OA-weighted average), for the Hyytiälä during campaigns in 2011 and 2014, during wood burning episodes in S. Vittore and Magadino and laboratory wood burning experiments (fresh emissions, 10h and 30h atmospherically aged), b) impact of NOx on fraction of C3-C7 and C17-C22 relative to C8-C12 compounds, c) ratio of pinic acid to MBTCA from LC-MS as a function of temperature.




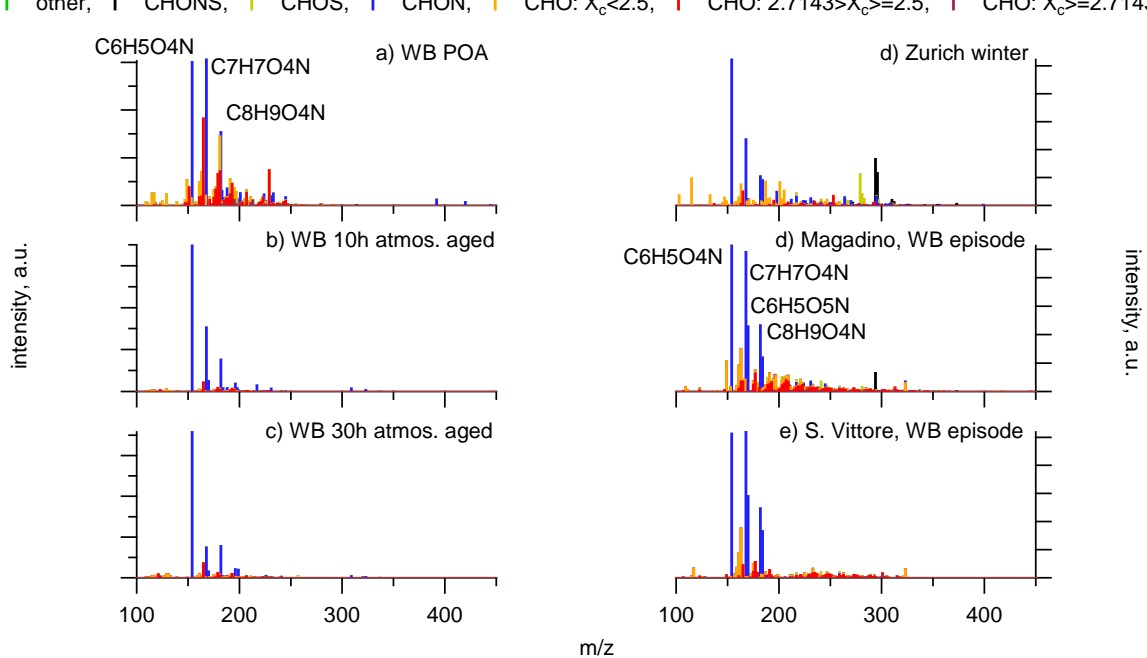

Figure 8: (-)ESI-ultra-high resolution mass spectra integrated to unit-mass resolution of wood burning laboratory experiments (a: primary emissions, b: emissions 10 hours atmospheric aging, c: emissions 30 hours atmospheric aging), and from winter wood burning episodes at Alpine valley sites (d: Magadino, e: S. Vittore). The signal at a nominal mass is separated by ion family (CHON, CHOS, CHONS, other) and the signal of CHO compounds by aromaticity (non-aromatic: $X_c > 2.5$, aromatic: $2.5 \leq X_c < 2.7143$, condensed aromatic: $X_c \geq 2.7143$). Peak assignments of dominant ions of selected UMR peaks are labelled as neutral compounds.





Figure 9: Van Krevelen diagrams of negative ion mode spectra (neutral composition) of smog chamber wood burning experiments (fresh emissions, after 10h of simulated atmospheric aging, and after 30h of simulated atmospheric aging) and of wood burning episodes at the Alpine valley sites Magadino and S. Vittore. Peaks are displayed as circles with their size reflecting log(intensity) and the color-code the molecular composition of the compound.



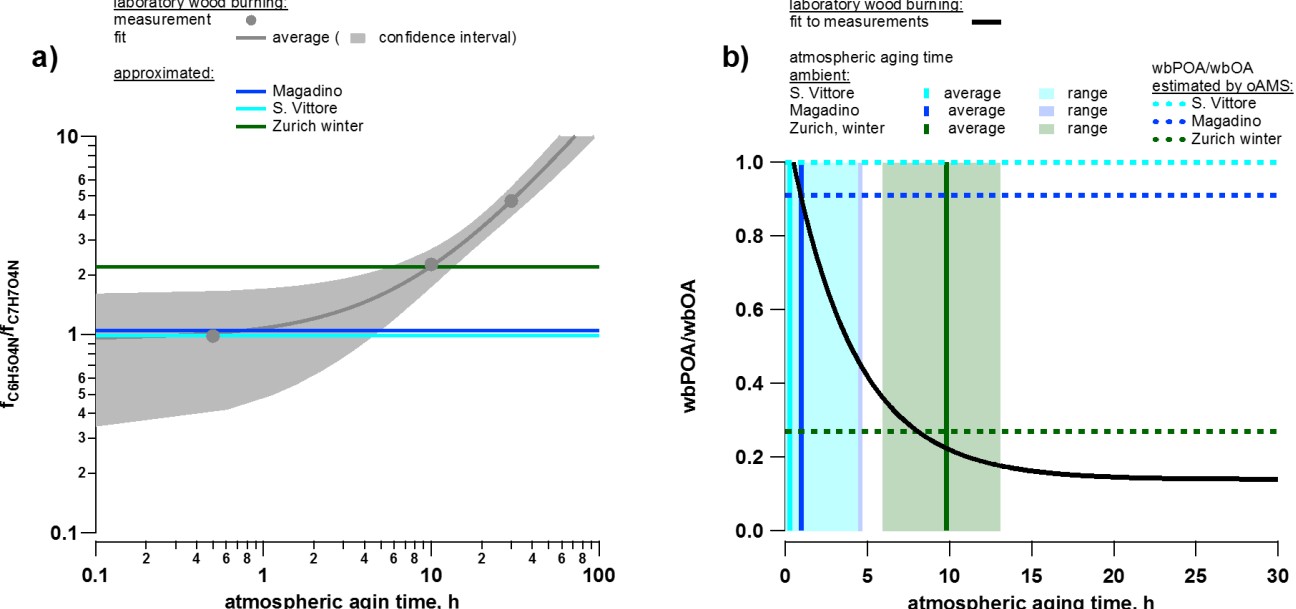

Figure 10: a) atmospheric aging parameterized as a function of the ratio of the fraction of total signal related to $C_6H_5O_4N$ ($f_{C6H5O4N}$) and $C_7H_7O_4N$ ($f_{C7H7O4N}$), $f_{C6H5O4N}/f_{C7H7O4N}$ for the wood burning smog chamber experiments and estimated atmospheric aging time of the wood burning emissions for the ambient analysis in Zurich (winter) and Magadino and S. Vittore.

5   b) Fraction of primary wood burning emissions to total wood burning OA (wbPOA/wbOA) parameterized as a function of the atmospheric aging time of the wood burning smog chamber experiments and estimated wbPOA/wbOA for the ambient analysis in Zurich (winter), Magadino, and S. Vittore.





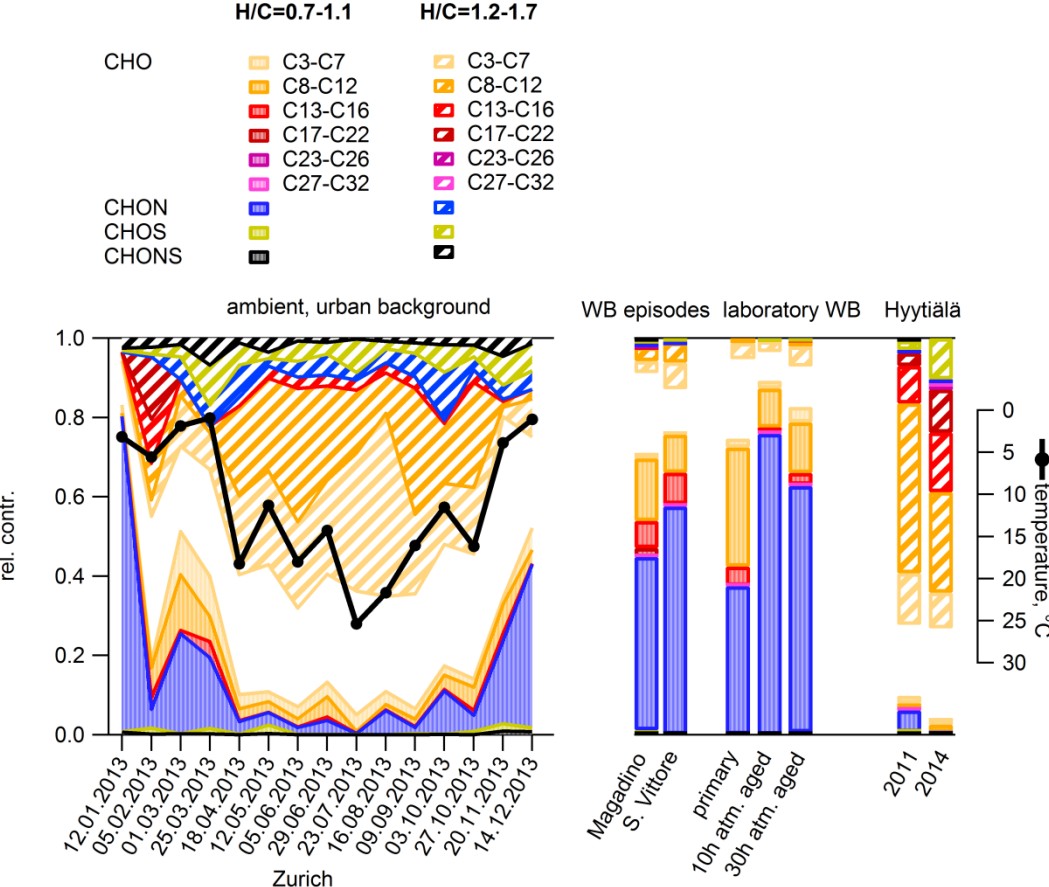

Figure 11: relative contributions of different compound classses to signal of compounds with $0.7 \leq H/C \leq 1.1$ as a surrogate for wood burning emissions and of compounds with $1.2 \leq H/C \leq 1.7$ as a surrogate for biogenic SOA.



Table 1: Concentrations of particulate and gaseous species and temperature.

| sample | | temp | particulate phase | | | | | | | gas phase | | | |
|---|---|---|---|---|---|---|---|---|---|---|---|---|---|
| | | | PM | OC | EC | NH₄⁺ | K⁺ | NO₃⁻ | SO₄²⁻ | NOx | O₃ | SO₂ | CO |
| | | °C | µg/m³ | | | | | | | ppb | | | ppm |
| **Ambient winter-time wood burning episode** | **Magadino (PM₁₀)** | 1.2 | 40.7 | 11.4 | 3.9 | 1.1 | 0.9 | 0.21 | 1.23 | 51 | 2 | 1.59 | - |
| | **S. Vittore (PM₁₀)** | 1.8 | 67.2 | 23.4 | 3.2 | 1.0 | 3.9 | 1.26 | 1.22 | 71 | - | - | 1.10 |
| **Ambient, Urban** | **Zurich, winter (PM₁₀)** | 2.8 | 23.4 | 3.0 | 0.6 | 2.9 | 0.3 | 6.9 | 3.4 | 31 | 14 | 0.85 | 0.34 |
| | **Zurich, summer (PM₁₀)** | 17 | 14.1 | 2.7 | 0.6 | 0.7 | 0.1 | 1.1 | 1.8 | 15 | 32 | 0.4 | 0.20 |
| **Ambient, remote** | **Hyytiälä, summer 2011 (PM₁)** | 15 | 3.6 | 2.7 | <0.1 | - | - | - | - | 0.6 | 20 | - | 0.1 |
| | **Hyytiälä, Summer 2014 (PM₁)** | 20 | 6.5 | 3.6 | 0.1 | - | - | - | - | 0.4 | 31 | - | 2 |





Table 2: Bulk properties of organic aerosol.

| sample | | $C_{bulk}$ | $(H/C)_{bulk}$ | $(O/C)_{bulk}$ | $(N/C)_{bulk}$ | $(S/C)_{bulk}$ |
|---|---|---|---|---|---|---|
| Laboratory wood burning | Primary | 8.3 | 0.99 | 0.50 | 0.057 | 0.0012 |
| | 10h aged | 7.2 | 0.93 | 0.61 | 0.114 | 0.0002 |
| | 30h aged | 7.6 | 0.94 | 0.61 | 0.091 | 0.0012 |
| Ambient wood burning episode | Magadino | 8.7 | 1.13 | 0.59 | 0.058 | 0.0065 |
| | S. Vittore | 8.1 | 1.09 | 0.61 | 0.072 | 0.0019 |
| Ambient, urban | Zurich winter | 7.8 | 1.24 | 0.74 | 0.076 | 0.0288 |
| | Zurich summer | 7.6 | 1.49 | 0.77 | 0.029 | 0.0253 |
| Ambient, remote | Hyytiälä (2011) | 9.7 | 1.48 | 0.50 | 0.025 | 0.0086 |
| | Hyytiälä (2014) | 11.3 | 1.50 | 0.63 | 0.001 | 0.0197 |