# Peer review of "Impact of anthropogenic and biogenic sources on the seasonal variation of the molecular composition of urban organic aerosols: a field and laboratory study using ultra-high resolution mass spectrometry"

_Atmospheric Chemistry and Physics, 2018_

## Referee Comment (RC1) · Anonymous Referee #1 · 27 Dec 2018

Daellenbach et al. propose a comprehensive characterization of the molecular composition of aerosols sampled at an urban site in Central Europe (Zurich, Switzerland). Chemical composition is retrieved using an ultra-high resolution mass spectrometry (Orbitrap) and further compare with aerosols sampled during wood burning emissions from Alpine valleys and chamber investigations of wood smoke. Finally, samples from the boreal forest were also used to evaluate the influence of biogenic emission in aerosol formation in Zurich. The results presented in this work are interesting and

provide important information on source apportionment of aerosol in Central Europe. The comparison lab and field data is particularly valuable. Overall, the interpretation and the results are well sustained. Therefore, I think the paper should be publishable after some comments are addressed.

General comments: page 2, lines 31-33: the authors mentioned that ESI coupled to a UHR-MS is a promising technique. It is now an established technique and cannot be classified as promising. Indeed many studies in atmospheric sciences and analytical chemistry have demonstrated the capabilities of the UHR-MS including the Orbitrap technology (commercialized by Thermo $\sim$ 15 years ago).

page 3, 1-3: Another major limitation of any offline technique compare to the AMS is the time resolution, which is worth mentioning.

page 3, 27-30: How many samples were analyzed? Different sizes (e.g., PM10, PM1,...) were chemically characterized and compared. However, the authors never mentioned the influence of the size, how would that impact the interpretation?

page 4, analytical procedure: The authors decided to use the Orbitrap in negative mode. Why didn't they explore the positive mode as well? As recently highlighted by e.g., Lin et al. (Anal Chem, 2018 10.1021/acs.analchem.8b02177) the positive mode can provide additional valuable information. The positive mode is generally less selective than the negative mode. Therefore for a global screening, both modes should be used.

page 4, lines 20-21. While replicate/triplicate measurements were performed the authors never mentioned the variability of their measurements. Screening analysis might bring large variability. Therefore, the authors should provide some statistical analysis in order to better validate their results/findings.

page 7, lines 21-23: As it is presented it is hard to see any correlation. Please provide the r or r2 for the different species to support the discussion (e.g., a table showing all

the r2 should be added). The authors mentioned that they measured the concentration of CO. How does CO correlate with other anthropogenic pollutants?

page 8, lines 4-15: It was already acknowledged by the authors that the relative contribution of a compound cannot be directly linked to its concentration(lines 1-3, page 3). However, it would be worth mentioning this point in this paragraph as it is an important aspect. Indeed, nitroaromatics are highly sensitive using ESI (-) but their large contribution to the MS doesn't imply that they are the most abundant species.

page 9, lines 18-19: Are the ratios (e.g., H/C or O/C) weighted by the area of the individual peak?

page 9, line 22: Accretion products imply aerosol processes (i.e., IUPAC definition). However, the chemistry describes by Berndt et al is a gas phase process. In addition, it is unlikely that these compounds arise from isoprene-RO2 + monoterpene-RO2 as isoprene concentration is very low in the Boreal forest and contributes overall to a small fraction of the OH and O3 reactivities (e.g., Hakola et al., 2012).

page 10, lines 1-5: Those products were also formed from the oxidation of isoprene (e.g., Surratt's group). Please check the literature and provide some information on the concentration of isoprene within the studied areas.

page 10, 3.4.2: The discussion of this paragraph is not consistent with the previous section. For instance, as it is written the authors suggest that the C4 & C5 compounds are formed from the aging of monoterpene-derived SOA but in the paragraph 3.4.2 they mention that the isoprene emissions are larger in Zurich than in Hyytiala, implying that isoprene chemistry plays a bigger role in Zurich. Please clarify and make the discussion more consistent.

page 13, lines 33-34: Based on the molecular signature of this group, can the authors propose a potential source? Could it be the VCP recently highlighted by McDonald et al. (2018, science)?

Figure 3b: Why is the dendrogram not symmetric? For instance, hyytiala 2011 vs hyytiala 2014 is different than hyytiala 2014 vs hyytiala 2011. It should not be like that, or should it be (if so, please explain)? In addition, the axes are not consistent compare to Figure 3a. Please revise Figure 3b to be consistent with Figure 3a.

Figure 5 (and S3) is hard to read. Please make all the graphs bigger. Another option would be to split the figure and have one figure for biogenic conditions with Zurich summer, Hyytiala 2011/2014 and possibly Zurich winter. Another figure will include wood burning experiments and episodes as well as Zurich winter.

Figure 7a is really hard to read and does not bring much information, as it is. It can be one separate figure and once again split between biogenic and wood burning SOA.

Figure 9 (and S4) doesn't include Zurich winter. Why?

technical comments: page 1, line 19: define OA page 2, line 22: Marseilles should be Marseille

---

## Referee Comment (RC2) · Anonymous Referee #2 · 29 Jan 2019

General comments

The authors present an analysis of Orbitrap ultra-high resolution mass spectra from organic aerosol (OA) filter samples collected over the course of 2013 in Zurich, Switzerland. These spectra are also compared to specific source spectra from laboratory wood burning experiments, and from ambient locations that are clearly influenced by a specific source (wood burning, biogenic emissions). The results show that summer OA in

[Figure]

Zurich is dominated by compounds of biogenic origin, whereas winter OA is dominated by compounds from wood burning emissions, and confirm the importance of non-fossil organic carbon for OA mass loadings in Central Europe.

This is a well and clearly written paper. Whereas the main message may not be completely new (it has been shown before by the same group that winter OA in Zurich is dominated by wood burning, and summer OA by biogenic secondary OA - the authors also refer to these studies), it is still of importance to corroborate it, as it has implications for air quality policy making. The ultra-high resolution spectra provide a new layer of information on the molecular composition of OA at this location. In addition to confirming the importance of wood burning and biogenic emissions for this location, it would also be interesting to use the detailed molecular information for improved estimates of OA health effects and/or physicochemical properties influencing climate effects.

I recommend this paper to be published after the following comments have been taken care of:

Specific comments

P. 1, l. 26: I suggest to add also the O/C ratio and average carbon number.

P. 3, l. 20 – 30: In Zurich, Magadino, and San Vittore, PM10 was measured, in Hyytiälä, PM1. The authors should explain if, and how these differences in size cut influence the measured chemical composition. The sampling time for the filters in Switzerland was 24 hours; what was the sampling time in Hyytiälä?

P. 3, l. 29 – 30: Why would temperature be the only parameter that can vary between years? Do the authors assume e.g. emissions to be the same, and why?

P. 4, l. 4: What kind of filters were used in the smog chamber experiments?

P. 4, l. 15: The authors should motivate their choice for using negative mode only.

P. 4, l. 18: Would it be possible for the differences in vaporizer temperature to produce artefacts? Please elaborate.

P. 7, l. 21: I realize this paragraph summarizes previous, already published studies, and is therefore not up for discussion here. However, it is not fully clear to me how WOOA and BBOA (Figure 1) are connected to the results presented here. WOOA is interpreted as being formed from anthropogenic VOC emissions – however, I am assuming emissions are similar between summer and winter. With the biogenic emissions becoming more important in the summer months I can see how the relative contributions WOOA become much less important in summer (Figure 1). What about the absolute concentrations, however? And if OA is really dominated by Wood Burning in winter, as the Orbitrap spectra suggest, then a large fraction of WOOA must be from Wood Burning as well. How come it correlates with NH4+? On p. 15, l. 5 - 7 the authors state that " a good correlation was observed between the relative signal contribution of the compounds with H/C between 0.7 and 1.1 and the relative contribution of the sum of BBOA and WOOA to OA" – does this refer to a correlation of time series? How does a potential mass closure look like? What are the compounds in the "white parts" of Figure 11? Meteorology (inversion episodes) play a major role as well in Zurich, especially during winter. How is that taken into account in the present study?

P. 7, l. 28 / Figure 2: What are the 9 / 6 summer / winter samples? Please give more details.

P. 8, l. 31 – 34, p. 9, l. 1 -3: It would have been interesting to compare to spectra not from the boreal forest, but from the temperate broadleaf forest dominating Central Europe. Would it be possible for the authors to add such a comparison?

P. 10, l. 16 – 19, p. 11, l. 4 - 8: Such ratios are highly dependent on plant species. Given the differences in biome between Hyytiälä and Zurich, I highly question if such statements can be made without further proof.

P. 14, l. 6 – 11: It was shown earlier that wood burning OA in Zurich is mostly of

regional origin, especially during inversion episodes (Mohr et al., ACP, 2011), which is in agreement with its longer lifetime compared to the Alpine valleys shown here. Since meteorology, and especially inversion episodes, play such an important role for winter air quality in Zurich, periods when there was such an episode should be marked in e.g. Figure 1.

P. 28, Figure 1: What is the reason for the increase in NOx emissions in the winter months, and the very high fraction of SC-OA in November?

Technical corrections

P. 5, equation 1: Define N

P. 11, 10: Influence on what? – Sentence should be rephrased.

P. 28, Fig. 1: It would improve readability of the graph if the sample days were just labelled "Jan, Feb. . .", and the dates were added in a table.

P. 32, Figure 5: This figure is hard to read and could potentially be left out.

P. 33, Figure 7a: What are the grey lines?

---

## Author Comment (AC1) · 12 Mar 2019

Daellenbach et al. propose a comprehensive characterization of the molecular composition of aerosols sampled at an urban site in Central Europe (Zurich, Switzerland). Chemical composition is retrieved using an ultra-high resolution mass spectrometry (Orbitrap) and further compare with aerosols sampled during wood burning emissions from Alpine valleys and chamber investigations of wood smoke. Finally, samples from the boreal forest were also used to evaluate the influence of biogenic emission in aerosol formation in Zurich. The results presented in this work are interesting and provide important information on source apportionment of aerosol in Central Europe. The comparison lab and field data is particularly valuable. Overall, the interpretation and the results are well sustained. Therefore, I think the paper should be publishable after some comments are addressed.

We thank the reviewer for the positive feedback and will answer the comments point-by-point below.

General comments:

page 2, lines 31-33: the authors mentioned that ESI coupled to a UHR-MS is a promising technique. It is now an established technique and cannot be classified as promising. Indeed many studies in atmospheric sciences and analytical chemistry have demonstrated the capabilities of the UHR-MS including the Orbitrap technology (commercialized by Thermo□15 years ago).

We reformulated the sentence to:

*"The use of soft ionization techniques, such as electrospray ionization coupled to ultra-high-resolution mass spectrometry (ESI-UHR-MS) is a powerful technique that may help bridging such existing gaps (Nizkorodov et al., 2011)."*

page 3, 1-3: Another major limitation of any offline technique compare to the AMS is the time resolution, which is worth mentioning.

We added this information, on page 3, line 10-12:

*"However, the typical number of samples analyzed remains very limited with low temporal resolution in comparison to online measurement techniques."*

page 3, 27-30: How many samples were analyzed? Different sizes (e.g., PM10, PM1,...) were chemically characterized and compared. However, the authors never mentioned the influence of the size, how would that impact the interpretation?

We performed the following Orbitrap analyses:

1) *15 PM10 filter samples from Zurich, Switzerland (24h sampling time), covering the entire year of 2013 (Daellenbach et al., 2017).*
2) *1 composite of 4 PM10 filter samples from Magadino, Switzerland (24h sampling time), highly influenced by biomass burning for wintertime residential heating.*
3) *1 composite of 4 PM10 filter samples from S. Vittore, Switzerland (24h sampling time), highly influenced by biomass burning for wintertime residential heating.*

4) 1 composite of 16 PM10 filter samples of fresh biomass burning emissions from laboratory experiments (30mins sampling time)
5) 1 composite of 11 PM10 filter samples of 10h equivalent atmospherically aged biomass burning emissions from laboratory experiments (30mins sampling time)
6) 1 composite of 15 PM10 filter samples of 30h equivalent atmospherically aged biomass burning emissions from laboratory experiments (30mins sampling time)

Additionally, we included 2 average spectra from Hyytiälä (PM1, summer 2011: 12h sampling time, summer 2014: 48 to 64h sampling time) representing biogenic SOA published in Kourtchev et al. (2016). Biogenic SOA is mostly in PM1 (Vlachou et al., 2018), thus comparing PM1 samples from Hyytiälä to PM10 samples from Zurich doesn't affect the conclusions related to biogenic SOA.

We added the following information in the new version of the manuscript:

"

The field samples at the 3 sites in Switzerland were daily collected in 2013 using a High Volume samplers (500 l min$^{-1}$, quartz-fiber filter, 14.7 cm filter diameter). Zurich is located on the northern Swiss plateau and the site is classified as urban background. The sites in Magadino and S. Vittore are in Alpine valleys in Southern Switzerland. While the 15 samples collected in Zurich were analyzed individually, the 4 samples from Magadino and 4 samples from S. Vittore were grouped and analyzed as composites (10.12.2013, 14.12.2013, 18.12.2013, 22.12.2013).

Samples from smog chamber experiments were analyzed to examine the composition of wood burning emissions from the stable flaming phase and their evolution with aging (Bruns et al., 2016; 2017). During these experiments, fresh emissions were first injected into a 6 m$^3$ Teflon smog chamber (Platt et al., 2013; Bruns et al., 2016, 2017). After 30 min of mixing, particles were sampled onto quartz-fiber filters (UV-lights off, sampling time 30 mins at ~30 l/min). Then, emissions were photochemically aged in the smog chamber, by injecting HONO at a flow rate of 1-2 l/min, which generates OH radicals upon photolysis. Samples were collected before aging (fresh) and at equivalent atmospheric aging times of 10 and 30 h (determined by the Barmet et al. (2012) method, assuming a winter time OH concentration of 10$^6$ molec cm$^{-3}$). All samples were grouped and measured as three composites: fresh wood burning emissions (16 samples), 10h equivalent atmospheric aging (11 samples) and 30h equivalent atmospheric aging (15 samples).

Further, we included 2 average spectra from the SMEAR-II station (Station for Measuring Ecosystem Atmosphere Relations, Hari and Kulmala 2005) at Hyytiälä (PM1, summer 2011: 12h sampling time, summer 2014: 48 to 64h sampling time, both quartz-fiber filters) previously published in Kourtchev et al. (2016). The SMEAR-II station (Station for Measuring Ecosystem Atmosphere Relations, Hari and Kulmala 2005) at Hyytiälä is a rural background site in Finland, strongly influenced by biogenic SOA (vegetation dominated by Scots pine and Norway spruce). The PM$_1$ aerosol was collected between 16 and 25 August 2011 and between 7 July and 4 August 2014 using a low volume sampler (35 l min$^{-1}$). Since biogenic SOA is mostly in PM1 (Vlachou et al., 2018), comparing PM1 samples from Hyytiälä to PM10 samples from Zurich does not affect the conclusions related to biogenic SOA.

*In addition, we collected samples from Frauenfeld (PM$_{10}$), Payerne (PM$_{10}$), Bern (PM$_{10}$), Zurich (PM$_{2.5}$) from the same year to complement the dataset with further chemical analyses (Section 2.3).*

*"*

page 4, analytical procedure: The authors decided to use the Orbitrap in negative mode. Why didn't they explore the positive mode as well? As recently highlighted by e.g., Lin et al. (Anal Chem, 2018 10.1021/acs.analchem.8b02177) the positive mode can provide additional valuable information. The positive mode is generally less selective than the negative mode. Therefore for a global screening, both modes should be used.

With the specific interest of this study in SOA sources and the large contribution of acids to SOA, we opted for analyzing the aerosol in the negative mode. However, in the light of recent publications we agree with the reviewer and will in future studies use both positive and negative mode analyses. We have added the following information:

*"Recent results show that the positive mode ESI, which is less selective, can provide additional valuable information, especially regarding fresh emissions (Lin et al., 2018). Here, to compare with previous results and as the main aim is to characterize SOA, we have focused on the negative mode ESI analysis."*

page 4, lines 20-21. While replicate/triplicate measurements were performed the authors never mentioned the variability of their measurements. Screening analysis might bring large variability. Therefore, the authors should provide some statistical analysis in order to better validate their results/findings.

We added an assessment of the variability among the replicate measurements in the supplementary information and mention it in the main text:

Text added in manuscript:

*"Two or three replicate measurements were conducted for each extract (variability assessed in Fig. S9), and field blank extracts were analyzed in the same way."*

Text added in the supplementary information:

***variability among replicate measurements***

*In order to estimate the relative error we performed replicate measurements of all samples (here computed as: $(x_{i,\max} - x_{i,\min})/x_{i,\text{avg}}$), with $x_{i,max}$, $x_{i,min}$, $x_{i,avg}$ being the maximum, minimum, and average peak (i) intensity measured for a respective sample. While the relative error varies considerably for a constant median peak intensity, overall typically the relative error of peaks ranges between 8 and 27%.*

[Figure]

*Figure S9: relative error as a function of the average signal intensity displayed for all samples and peaks.*

page 7, lines 21-23: As it is presented it is hard to see any correlation. Please provide the r or r2 for the different species to support the discussion (e.g., a table showing all the r2 should be added). The authors mentioned that they measured the concentration of CO. How does CO correlate with other anthropogenic pollutants?

In the present study we analyzed 15 samples from Zurich from a larger study with 91 samples from the same site. This paragraph introduces the bigger picture and the correlations are presented in Daellenbach et al. (2017). We now state the correlations and cite the publication more explicitly:

*"WOOA correlated with anthropogenically-influenced inorganic ions like $NH_4^+$ and was for this reason interpreted as being formed from anthropogenic VOC emissions. SOOA in contrast showed a positive non-linear relation to temperature, consistent with the temperature driven enhancement of biogenic terpene emissions (Daellenbach et al., 2017, for entire dataset from Zurich $R_p^2(WOOA,NH_4^+,n=90)=0.79$, $R_s(SOOA,temp,n=91)=0.65$)."*

page 8, lines 4-15: It was already acknowledged by the authors that the relative contribution of a compound cannot be directly linked to its concentration(lines 1-3, page 3). However, it would be worth mentioning this point in this paragraph as it is an important aspect. Indeed, nitroaromatics are highly sensitive using ESI (-) but their large contribution to the MS doesn't imply that they are the most abundant species.

We have added the following content to the manuscript:

*"We present the average summer (T>11°C, 18.04.2013, 12.05.2013, 05.06.2013, 29.06.2013, 23.07.2013, 16.08.2013, 09.09.2013, 03.10.2013, 27.10.2013) and winter (T<6°C, 12.01.2013, 05.02.2013, 01.03.2013, 25.03.2013, 20.11.2013, 14.12.2013) spectra from (-)ESI-UHR-MS at the urban background site in Zurich (mass spectral signature and van Krevelen diagrams in Fig. 2). We note that peak intensities, $x_i$, are not directly linked to concentrations and only relative differences can be interpreted. The summer and winter*

*average spectra exhibited a strong seasonal difference. During summer, peaks related to compounds only containing carbon, hydrogen and oxygen (CHO) dominated the spectrum. The majority of these compounds had a ratio H/C around 1.5 and O/C between 0.4 and 1.4 (Fig. 2). These compounds were either absent or had a much lower intensity during winter. "*

page 9, lines 18-19: Are the ratios (e.g., H/C or O/C) weighted by the area of the individual peak?

Yes they are weighted by the peak intensity. We would like to refer to section 2.2.3 in the manuscript where we describe the computation of the properties in detail:

*All properties, molar ratios, and chemical formulae presented in this manuscript refer to neutral molecules. Literature data was additionally also filtered with criterion (2) for comparability. Bulk elemental ratios (H/C, O/C, N/C and S/C) and the number of carbons of the organic aerosol were computed as follows (Nizkorodov et al., 2011, Bateman et al., 2012):*

$$(O/C)_{bulk} = \sum_i x_i * N_{O,i} / \sum_i x_i * N_{C,i} \tag{2}$$

$$(H/C)_{bulk} = \sum_i x_i * N_{H,i} / \sum_i x_i * N_{C,i} \tag{3}$$

$$(N/C)_{bulk} = \sum_i x_i * N_{N,i} / \sum_i x_i * N_{C,i} \tag{4}$$

$$(S/C)_{bulk} = \sum_i x_i * N_{S,i} / \sum_i x_i * N_{C,i} \tag{5}$$

$$C_{bulk} = \sum_i x_i * N_{C,i} / \sum_i x_i \tag{6}$$

page 9, line 22: Accretion products imply aerosol processes (i.e., IUPAC definition). However, the chemistry describes by Berndt et al is a gas phase process. In addition, it is unlikely that these compounds arise from isoprene-RO2 + monoterpene-RO2 as isoprene concentration is very low in the Boreal forest and contributes overall to a small fraction of the OH and O3 reactivities (e.g., Hakola et al., 2012).

We adapted the paragraph in question to:

*"...*

*The C13-C16 compounds are thought to consist mainly of sesquiterpene oxidation products, but may also be produced through reactions of monoterpene and isoprene $RO_2$ radicals (Berndt et al., 2018), which is less probable in the boreal forest due to the low isoprene concentrations.*

*"*

page 10, lines 1-5: Those products were also formed from the oxidation of isoprene (e.g., Surratt's group). Please check the literature and provide some information on the concentration of isoprene within the studied areas.

We added that also isoprene can form such small compounds and refer to section 3.4.2 where we studied the isoprene/monoterpene emission ratios in the region around Hyytiälä and Zurich (Fig. S2):

"*Some of these compounds were related to OH radical induced atmospheric aging of monoterpene SOA, especially at high NOx conditions, in ambient as well as in laboratory experiments (Zhang et al., 2018; Mutzel et al., 2015) but could also originate from other biogenic precursors such as isoprene (see section 3.4.2).* "

We highlight the information on page 10, L25-27:

"

*Modelled biogenic emissions showed a higher isoprene (ISO, $C_5H_8$) to monoterpene (MT, $C_{10}H_{16}$) ratio in Switzerland than in Finland (Fig. S2, Jiang et al., 2018). The higher ISO/MT ratio in BVOC emissions in Zurich could contribute to the higher C3-C7 CHO compound contribution at this site (see above, Fig. 4, 5, 7).*

"

Supplementary Figure S9:

[Figure]

*Figure S2: biogenic emissions of isoprene (ISO), monoterpene (MT), and sesquiterpenes (SQT) displayed as ratios SQT/MT and ISO/MT for the area (approx. 450 km x 450 km) surrounding Zurich, Switzerland, and Hyytiälä, Finland, calculated for summer 2011 using the MEGAN biogenic emission model (Jiang et al., 2018)*

page 10, 3.4.2: The discussion of this paragraph is not consistent with the previous section. For instance, as it is written the authors suggest that the C4 & C5 compounds are formed from the aging of monoterpene-derived SOA but in the paragraph 3.4.2 they mention that the isoprene emissions are larger in Zurich than in Hyytiala, implying that isoprene chemistry plays a bigger role in Zurich. Please clarify and make the discussion more consistent.

As detailed in the manuscript C4-C5 can be enhanced in Zurich compared to Hyytiälä for different reasons: 1) higher NOx concentrations (Zurich>Hyytiälä) lead to enhanced fragmentation of monoterpenes (detailed in 3.4.4), and 2) higher isoprene/monoterpene emission ratio in Zurich than Hyytiälä (detailed in section 3.4.2).

We refer to the discussion in the manuscript.

Section 3.4.1, page 10, L16-21:

*Meanwhile, small molecules such as $C_4H_6O_5$ (possibly related to malonic acid) and $C_5H_8O_5$ (possibly related to hydroxyglutaric acid) exhibited a higher fractional contribution in Zurich during summer than in Hyytiälä 2014 (Fig. 4, 6, 7, 8). Some of these compounds were related to OH radical induced atmospheric aging of monoterpene SOA, especially at high NOx conditions, in ambient as well as in laboratory experiments (Zhang et al., 2018; Mutzel et al., 2015) but could also originate from other biogenic precursors such as isoprene (see section 3.4.2). In the following, we will discuss the possible reasons for the differences.*

Section 3.4.2, page 10, L27-28:

*The higher ISO/MT ratio in BVOC emissions in Zurich could contribute to the higher C3-C7 CHO compound contribution at this site (see above, Fig. 4, 5, 7).*

Section 3.4.3, page 11, L20-24:

*The increase in the proportion of smaller compounds (C3-C7) occurs despite their increasingly higher evaporation rates. This could be related to a higher fraction of $1^{st}$ generation products residing in the gas-phase where they are prone to further oxidation, possibly also promoting fragmentation. Since the average temperature in Zurich during summer is 17°C (average $T_{max}=21°C$) this would partially explain the enhancement of the fraction of lower molecular weight compounds (C3-C7) compared to Hyytiälä.*

Section 3.4.4, page 11, L26-31:

*While laboratory monoterpene experiments show an important influence of functionalized monomeric oxidation products, ambient measurements have revealed an enhancement of fragmentation over functionalized products with increasing NOx concentrations (Zhang et al., 2018). Fragmentation products of $RO_2$ + NO reactions and subsequent autooxidation could explain such observation. Since we observe a similar behavior (Fig. 7b) in this study, the higher (C3-C7)/(C8-C12) ratio in summertime Zurich than in Hyytiälä can be related to enhanced NOx concentrations at the urban site (NOx summertime Zurich: 15 ppb, Hyytiälä: 0.5 ppb).*

page 13, lines 33-34: Based on the molecular signature of this group, can the authors propose a potential source? Could it be the VCP recently highlighted by McDonald et al. (2018, science)?

The question refers to the unexplained compound class characterized by carbon numbers between 9 and 12 and H/C between 1.5 and 2.0. Volatile chemical products, VCP, might contribute to the observed unexplained compound class. However, in absence of VCP laboratory aging experiments to compare our ambient SOA signatures to, we are unable to hypothesize on the origin of these compounds.

Figure 3b: Why is the dendrogram not symmetric? For instance, hyytiala 2011 vs hyytiala 2014 is different than hyytiala 2014 vs hyytiala 2011. It should not be like that, or should it be (if so, please explain)? In addition, the axes are not consistent compare to Figure 3a. Please revise Figure 3b to be consistent with Figure 3a.

Figure 3a presents a 2D clustergram and is sorted in both dimension and figure 3b only in one dimension, a 1D clustergram. The cells in Figure represent the fraction of peaks that a certain spectrum has in common with the sample indicated on the right axis of the figure.

Given 2 samples i and ii, the correlation R is the same for $R(i,ii)$ and $R(ii,i)$. The fraction of peaks that a sample $i$ has in common with $ii$ (number of peaks $k$) $k(i \cap ii)/k(i)$ is not the same as $k(ii \cap i)/k(ii)$. Therefore, only a 1D analysis can be performed.

We added more information in Section 2.4:

"

*In approach B, we computed the number of peaks k that a sample i had in common with another sample ii normalized to the total number of peaks detected in sample i (k(i ∩ ii)/ k(i))*

"

We now added more information to the figure caption apply the same sorting also to the other dimension for easier readability.

[Figure]

Figure 3: a) correlation matrix of mass spectra sorted by hierarchical cluster analysis also depicting the similarity as dendrograms, b) number of common peaks of a sample with the sample indicated on the y-axis normalized to the total number of peaks of the respective sample sorted by hierarchical cluster analysis also depicting the similarity as dendrograms.

"

Figure 5 (and S3) is hard to read. Please make all the graphs bigger. Another option would be to split the figure and have one figure for biogenic conditions with Zurich summer, Hyytiala 2011/2014 and possibly Zurich winter. Another figure will include wood burning experiments and episodes as well as Zurich winter.

The figures cannot be enlarged within the panel but we agree with the reviewer that the subpanels need to be well visible. This figure should be a full page figure in the final manuscript (we adapted the manuscript in this sense).

[Figure]

Figure 5: Probability density functions (pdf) and contributions of different molecule families to H/C for all molecules (neutral composition based on (-)ESI-ultra-high resolution mass spectra), molecules with 5 to 8 carbon atoms, and 9 to 12 carbon atoms for the ambient samples collected in Zurich, Magadino, S. Vittore, and wood burning smog chamber experiments. The area of the histograms is proportional to the percentage of the total signal explained for each data set.

Figure 7a is really hard to read and does not bring much information, as it is. It can be one separate figure and once again split between biogenic and wood burning SOA.

For a better readability we split Figure 7a from 7b, c, and d:

"

[Figure]

Figure 7: Ultra-high-resolution mass spectra of CHO compounds integrated to unit-mass resolution in the negative mode for the organic aerosol for Zurich during summer ($T$>11°C) and winter ($T$<6°C) 2013 ($PM_{10}$, OA-weighted average), for the Hyytiälä during campaigns in 2011 and 2014, during wood burning episodes in S. Vittore and Magadino and laboratory wood burning experiments (fresh emissions, 10h and 30h atmospherically aged),

[Figure]

Figure 8: impact of NOx on fraction of (a) C3-C7 and (b) C17-C22 relative to C8-C12 compounds, c) ratio of pinic acid to MBTCA from LC-MS as a function of temperature.

Figure 9 (and S4) doesn't include Zurich winter. Why?

Equivalent figures are presented in Fig. 2 and Fig. S5.

technical comments:

page 1, line 19: define OA page 2, line 22: Marseilles should be Marseille

We corrected the mistake.

---

## Author Comment (AC2) · 12 Mar 2019

General comments

The authors present an analysis of Orbitrap ultra-high resolution mass spectra from organic aerosol (OA) filter samples collected over the course of 2013 in Zurich, Switzerland. These spectra are also compared to specific source spectra from laboratory wood burning experiments, and from ambient locations that are clearly influenced by a specific source (wood burning, biogenic emissions). The results show that summer OA in Zurich is dominated by compounds of biogenic origin, whereas winter OA is dominated by compounds from wood burning emissions, and confirm the importance of non-fossil organic carbon for OA mass loadings in Central Europe. This is a well and clearly written paper. Whereas the main message may not be completely new (it has been shown before by the same group that winter OA in Zurich is dominated by wood burning, and summer OA by biogenic secondary OA - the authors also refer to these studies), it is still of importance to corroborate it, as it has implications for air quality policy making. The ultra-high resolution spectra provide a new layer of information on the molecular composition of OA at this location. In addition to confirming the importance of wood burning and biogenic emissions for this location, it would also be interesting to use the detailed molecular information for improved estimates of OA health effects and/or physicochemical properties influencing climate effects.

I recommend this paper to be published after the following comments have been taken care of:

We thank the reviewer for the positive feedback and will answer the comments point-by-point below.

Specific comments

P. 1, l. 26: I suggest to add also the O/C ratio and average carbon number.

We added this information to the abstract:

"

Samples from Zurich during summer are similar to those collected in Hyytiälä, predominantly impacted by oxygenated compounds with an H/C ratio of 1.5, indicating the importance of biogenic precursors for SOA formation at this location (summertime Zurich: carbon number 7.6, O/C 0.74, Hyytiälä: carbon number 10.5 O/C 0.57).

"

P. 3, l. 20 – 30: In Zurich, Magadino, and San Vittore, PM10 was measured, in Hyytiälä, PM1. The authors should explain if, and how these differences in size cut influence the measured chemical composition. The sampling time for the filters in Switzerland was 24 hours; what was the sampling time in Hyytiälä?

A similar comment was raised by referee 1, we have copied below our response to referee 1:

We performed the following analyses:

  1) 15 PM10 filter samples from Zurich, Switzerland (24h sampling time), covering the entire year of 2013 (Daellenbach et al., 2018).

2) *1 composite of 4 PM10 filter samples from Magadino, Switzerland (24h sampling time), highly influenced by biomass burning for wintertime residential heating.*

4) *1 composite of 16 PM10 filter samples of fresh biomass burning emissions from laboratory experiments (30mins sampling time)*

Additionally, we included 2 average spectra from Hyytiälä (PM1, summer 2011: 12h sampling time, summer 2014: 48 to 64h sampling time) representing biogenic SOA published in Kourtchev et al. (2016). Biogenic SOA is mostly in PM1 (Vlachou et al., 2018), thus comparing PM1 samples from Hyytiälä to PM10 samples from Zurich doesn't affect the conclusions related to biogenic SOA.

We added the following information in the new version of the manuscript:

"

*The field samples at the 3 sites in Switzerland were daily collected in 2013 using a High Volume samplers (500 l min$^{-1}$, quartz-fiber filter, 14.7 cm filter diameter). Zurich is located on the northern Swiss plateau and the site is classified as urban background. The sites in Magadino and S. Vittore are in Alpine valleys in Southern Switzerland. While the 15 samples collected in Zurich were analyzed individually, the 4 samples from Magadino and 4 samples from S. Vittore were grouped and analyzed as composites (10.12.2013, 14.12.2013, 18.12.2013, 22.12.2013).*

*Samples from smog chamber experiments were analyzed to examine the composition of wood burning emissions from the stable flaming phase and their evolution with aging (Bruns et al., 2016; 2017). During these experiments, fresh emissions were first injected into a 6 m$^3$ Teflon smog chamber (Platt et al., 2013; Bruns et al., 2016, 2017). After 30 min of mixing, particles were sampled onto quartz-fiber filters (UV-lights off, sampling time 30 mins at ~30 l/min). Then, emissions were photochemically aged in the smog chamber, by injecting HONO at a flow rate of 1-2 l/min, which generates OH radicals upon photolysis. Samples were collected before aging (fresh) and at equivalent atmospheric aging times of 10 and 30 h (determined by the Barmet et al. (2012) method, assuming a winter time OH concentration of 10$^6$ molec cm$^{-3}$). All samples were grouped and measured as three composites: fresh wood burning emissions (16 samples), 10h equivalent atmospheric aging (11 samples) and 30h equivalent atmospheric aging (15 samples).*

*Further, we included 2 average spectra from the SMEAR-II station (Station for Measuring Ecosystem Atmosphere Relations, Hari and Kulmala 2005) at Hyytiälä (PM1, summer 2011: 12h sampling time, summer 2014: 48 to 64h sampling time, both quartz-fiber filters) previously published in Kourtchev et al. (2016). The SMEAR-II station (Station for Measuring Ecosystem Atmosphere Relations, Hari and Kulmala 2005) at Hyytiälä is a rural background site in Finland, strongly influenced by biogenic SOA (vegetation dominated by Scots pine and Norway spruce). The PM$_1$ aerosol was collected between 16 and 25 August 2011 and between 7 July and*

*4 August 2014 using a low volume sampler (35 l min⁻¹). Since biogenic SOA is mostly in PM1 (Vlachou et al., 2018), comparing PM1 samples from Hyytiälä to PM10 samples from Zurich does not affect the conclusions related to biogenic SOA.*

*In addition, we collected samples from Frauenfeld (PM₁₀), Payerne (PM₁₀), Bern (PM₁₀), Zurich (PM₂.₅) from the same year to complement the dataset with further chemical analyses (Section 2.3).*

*"*

P. 3, l. 29 – 30: Why would temperature be the only parameter that can vary between years? Do the authors assume e.g. emissions to be the same, and why?

We removed this sentence.

P. 4, l. 4: What kind of filters were used in the smog chamber experiments?

We used quartz fiber samples as for all other sites/experiments. This information is now added to the manuscript:

*"...*

*Samples from smog chamber experiments were analyzed to examine the composition of wood burning emissions from the stable flaming phase and their evolution with aging (Bruns et al., 2016; 2017). During these experiments, fresh emissions were first injected into a 6 m³ Teflon smog chamber (Platt et al., 2013; Bruns et al., 2016, 2017). After 30 min of mixing, particles were sampled onto quartz-fiber filters (UV-lights off, sampling time 30 mins at ~30 l/min). Then, emissions were photochemically aged in the smog chamber, by injecting HONO at a flow rate of 1-2 l/min, which generates OH radicals upon photolysis. Samples were collected before aging (fresh) and at equivalent atmospheric aging times of 10 and 30 h (determined by the Barmet et al. (2012) method, assuming a winter time OH concentration of 10⁶ molec cm⁻³). All samples were grouped and measured as three composites: fresh wood burning emissions (16 samples), 10h equivalent atmospheric aging (11 samples) and 30h equivalent atmospheric aging (15 samples).*

*...*"

P. 4, l. 15: The authors should motivate their choice for using negative mode only.

A similar comment was raised by referee 1, we have copied below our response to referee 1:

With the specific interest of this study in SOA's sources and the large contribution of acids to SOA, we opted for analyzing the aerosol in the negative mode. However, in the light of recent publications we agree with the reviewer and will in future studies use both positive and negative mode analyses. We have added the following information:

*"Recent results show that the positive mode ESI, which is less selective, can provide additional valuable information, especially regarding fresh emissions (Lin et). Here, to compare with previous results and as the main aim is to characterize SOA, we have focused on the negative mode ESI analysis."*

P. 4, l. 18: Would it be possible for the differences in vaporizer temperature to produce artefacts? Please elaborate.

We added the following in formation to the supplementary information:

"

*We investigated the possibility of artefacts induced by differences in vaporizer temperature by measuring 2 samples at the two used vaporizer temperatures (230°C-254°C). We found that differences induced by the temperature change were not statistically higher than our repeatability and were clearly lower than differences observed between samples. This suggests that temperature induced fragmentation artefacts are not a major driver of the observed chemical composition. The comparison between the two settings is shown in Figure S8.*

[Figure]

*Figure S8: Comparison of mass spectra recorded with 2 different instrumental settings vaporizer temperature 230°C and 254°C) for (a) a sample representing wood burning smoke (30h atmospherically aged) and (b) a sample representing wood burning smoke (10h) aged); (c) a comparison of the 2 samples measured with the same settings (vaporizer temperature 230°C). Measurements are displayed as black dots, 1:1 line as solid grey line, 1:2 and 2:1 line as strikethrough line.*

"

P. 7, l. 21: I realize this paragraph summarizes previous, already published studies, and is therefore not up for discussion here. However, it is not fully clear to me how WOOA and BBOA (Figure 1) are connected to the results presented here. WOOA is interpreted as being formed from anthropogenic VOC emissions – however, I am assuming emissions are similar between summer and winter. With the biogenic emissions becoming more important in the summer months I can see how the relative contributions WOOA become much less important in summer (Figure 1). What about the absolute concentrations, however?

Anthropogenic VOC emissions are from different sources in winter than summer. A strong additional wintertime source is residential heating such as wood burning.

Absolute concentrations of WOOA are much lower in summer than winter (Daellenbach et al., 2017.)

And if OA is really dominated by Wood Burning in winter, as the Orbitrap spectra suggest, then a large fraction of WOOA must be from Wood Burning as well. How come it correlates with NH4+?

$NH_4^+$ is a tracer for regionally transported aged emission from continental Europe (Zotter et al., 2014). $NH_3$ acts as neutralizing base for organic acids which explains the observed correlation between $NH_4^+$ and WOOA.

On p. 15, l. 5 - 7 the authors state that " a good correlation was observed between the relative signal contribution of the compounds with H/C between 0.7 and 1.1 and the relative contribution of the sum of BBOA and WOOA to OA" – does this refer to a correlation of time series?

Yes, here we compare time series, this information is now also added in the maintext at both instances:

*"In Zurich, a good correlation was observed between the time series of the relative signal contribution of compounds with H/C between 0.7 and 1.1 and the relative contribution of the sum of BBOA and WOOA to OA ($R^2$=0.59, p<0.001), with lower correlations with either BBOA alone ($R^2$=0.29, p=0.02) or WOOA alone ($R^2$=0.37, p<0.01)."*

*"In Zurich, a good correlation was observed between the time series of the the relative signal contribution of the compounds with H/C between 1.2 and 1.7 and the relative contribution of SOOA to OA ($R^2$=0.57, p<0.01)."*

P. 7, l. 28 / Figure 2: What are the 9 / 6 summer / winter samples? Please give more details.

We state now explicitly which samples belong to which category:

"

*We present the average summer (T>11°C, 18.04.2013, 12.05.2013, 05.06.2013, 29.06.2013, 23.07.2013, 16.08.2013, 09.09.2013, 03.10.2013, 27.10.2013) and winter (T<6°C, 12.01.2013, 05.02.2013, 01.03.2013, 25.03.2013, 20.11.2013, 14.12.2013) spectra from (-)ESI-UHR-MS at the urban background site in Zurich (mass spectral signature and van Krevelen diagrams in Fig. 2). We note that peak intensities, $x_i$, are not directly linked to concentrations and only relative differences can be interpreted. The summer and winter average spectra exhibited a strong seasonal difference. During summer, peaks related to compounds only containing carbon, hydrogen and oxygen (CHO) dominated the spectrum. The majority of these compounds had a ratio H/C around 1.5 and O/C between 0.4 and 1.4 (Fig. 2). These compounds were either absent or had a much lower intensity during winter.*

"

How does a potential mass closure look like?

Since we do not quantify the Orbitrap measurements and only look at peak intensities, we cannot attempt a mass closure.

What are the compounds in the "white parts" of Figure 11?

We couldn't relate these compounds neither to wood burning emissions nor to biogenic SOA. A large contribution stems from compounds that contain sulfur and/or nitrogen (see Figure S7). In Hyytiälä $SO_2$ and $NO_x$ levels are considerably lower than in Zurich and we didn't add $SO_2$ during the smog chamber experiments which might contribute to this observation.

Meteorology (inversion episodes) play a major role as well in Zurich, especially during winter. How is that taken into account in the present study?

Thermal inversion is indeed important for the absolute concentrations of the pollutants in winter, but potentially to a lesser extent for the composition, driven by the exposure to oxidants and the emission composition. The latter is the focus of the paper. We do not mean that inversion does not occur in Zurich, but that atmospheric inversion, coupled with high emissions and the valley topography at Magadino results in higher concentrations at this location.

P. 8, l. 31 – 34, p. 9, l. 1 -3: It would have been interesting to compare to spectra not from the boreal forest, but from the temperate broadleaf forest dominating Central Europe. Would it be possible for the authors to add such a comparison?

We agree with the reviewer that a comparison to biogenic SOA from similar vegetation as around Zurich would be ideal. However, we could not find an environment representing biogenic SOA better than Hyytiälä. Most places in central Europe are far less remote and more strongly impacted by anthropogenic VOC emissions. Therefore, we opted to compare to Hyytiälä.

P. 10, l. 16 – 19, p. 11, l. 4 - 8: Such ratios are highly dependent on plant species. Given the differences in biome between Hyytiälä and Zurich, I highly question if such statements can be made without further proof.

We agree with the reviewer that differences in biome between Hyytiälä and Zurich are important for SQT/MT, ISO/MT emission ratios and the contribution of C3-C7 compounds relative to larger compounds in biogenic SOA. For that reason, we examine the differences in ISO/MT and SQT/MT emission ratios in the regions around Hyytiälä and Zurich (Fig. S2) showing that ISO/MT is clearly enhanced in Zurich compared to Hyytiälä which reflects differences in biome (pinus silvestris in Hyytiälä: ISO/MT=0 µg g$^{-1}$h$^{-1}$/5 µg g$^{-1}$h$^{-1}$, picea abies in Switzerland: ISO/MT=1 µg g$^{-1}$h$^{-1}$/2.5 µg g$^{-1}$h$^{-1}$, Steinbrecher et al., 2009). However, as shown by Zhao et al. (2017) also biotic stress can impact the emission ratios of trees and as shown by Kourtchev et al. (2016) temperature seems to be important for the contribution of C3-C7 compounds to biogenic SOA. Therefore, we think it is important to point out these possibilities too.

We refer to the section in the maintext where we discuss differences in ISO/MT and SQT/MT emission ratios in Hyytiälä and Zurich:

The composition of BVOC emissions depends on various parameters such as vegetation type and temperature. While many BVOCs lead to the formation of oxidation products characterized by H/C ~1.5, $C_{bulk}$

depends on the size of the carbon backbone of the initially emitted precursor and the degree of accretion. Thus, the composition of the BVOC emissions has an impact on $C_{bulk}$. Modelled biogenic emissions showed a higher isoprene (ISO, $C_5H_8$) to monoterpene (MT, $C_{10}H_{16}$) ratio in Switzerland than in Finland (Fig. S2, Jiang et al., 2018). The higher ISO/MT ratio in BVOC emissions in Zurich could contribute to the higher C3-C7 CHO compound contribution at this site (see above, Fig. 4, 5, 7, 8). SQT/MT did not show a clear difference between Finland and Switzerland and is therefore not expected to be the reason for the observed enhanced abundance of C13-C17 compounds in Hyytiälä compared to summertime Zurich (however, see the NOx discussion in the next section).

We adapted the main text so that it is clear that we speak about the composition of biogenic SOA not biogenic vapors on page 11, L19-24:

"

*Kourtchev et al. (2016) observed an increasing fraction of smaller molecules (C3-C7) to the total observed signal from biogenic SOA at higher temperatures. The increase in the proportion of smaller compounds (C3-C7) occurs despite their increasingly higher evaporation rates. This could be related to a higher fraction of 1$^{st}$ generation products residing in the gas-phase where they are prone to further oxidation, possibly also promoting fragmentation. Since the average temperature in Zurich during summer is 17°C (average $T_{max}=21°C$) this would partially explain the enhancement of the fraction of lower molecular weight compounds (C3-C7) compared to Hyytiälä.*

"

P. 14, l. 6 – 11: It was shown earlier that wood burning OA in Zurich is mostly of regional origin, especially during inversion episodes (Mohr et al., ACP, 2011), which is in agreement with its longer lifetime compared to the Alpine valleys shown here. Since meteorology, and especially inversion episodes, play such an important role for winter air quality in Zurich, periods when there was such an episode should be marked in e.g. Figure 1.

For our case, the OA concentrations during winter remain moderate and are not especially higher than the rest of year (see figure 1). Therefore, we cannot really identify extreme inversion episodes in Zurich during winter. As mentioned earlier, the focus of the study is not to identify the effect of meteorology on the particle composition, but rather identify the chemical composition and the main sources of SOA during winter and summer.

P. 28, Figure 1: What is the reason for the increase in NOx emissions in the winter months, and the very high fraction of SC-OA in November?

The relative contribution of SCOA is high, though the absolute OA concentration is low. Overall, SC-OA is not especially high.

Technical corrections

P. 5, equation 1: Define N

This information is now added to the paragraph:

"

*Molecular assignments have to be consistent with a neutral formula with a positive integer double bond equivalent, (DBE, for any chemical formula $C_{N_C}H_{N_H}O_{N_O}N_{N_N}S_{N_S}$, $N_C$, $N_H$, $N_O$, $N_N$, and $N_S$ represent the number of carbon, hydrogen, oxygen, nitrogen, and sulfur atoms):*

*"*

P. 11, 10: Influence on what? – Sentence should be rephrased.

We adapted this sentence to:

*"...*

*While laboratory monoterpene experiments show a large contribution of functionalized monomeric oxidation products to SOA, ambient measurements have revealed an enhancement of fragmentation over functionalized products with increasing NOx concentrations (Zhang et al., 2018).*

*..."*

P. 28, Fig. 1: It would improve readability of the graph if the sample days were just

labelled "Jan, Feb. . .", and the dates were added in a table.

Since we have several samples for certain month, we unfortunately cannot call the samples by month only. Therefore, we have to keep the figure as it currently is.

P. 32, Figure 5: This figure is hard to read and could potentially be left out.

This figure shows what chemical components drive the variability in Figure 4 (not only H/C but also carbon number). Thus we would like to keep it in the manuscript.

P. 33, Figure 7a: What are the grey lines?

We agree that the lines confuse the reader and thus we removed them from the figure.

References:

Steinbrecher et al., Intra- and inter-annual variability of VOC emissions from natural and semi-natural vegetation in Europe and neighbouring countries, Atmos. Environ., 43, 1380-1391, doi: 10.1016/j.atmosenv.2008.09.072, 2009.